# In-Training Defenses Against Emergent Misalignment in Language Models

**David Kaczér** [1 2]  **Magnus Jørgenvåg** [1]  **Clemens Vetter** [1]  **Esha Afzal** [3]  **Robin Haselhorst** [1 3]  **Lucie Flek** [1 2]
**Florian Mai** [1 2]

## Abstract

Fine-tuning lets practitioners repurpose aligned large language models (LLMs) for new domains, yet recent work reveals emergent misalignment (EM): Even a small, domain-specific fine-tune can induce harmful behaviors far outside the target domain. Even in the case where model weights are hidden behind a fine-tuning API, this gives attackers inadvertent access to a broadly misaligned model in a way that can be hard to detect from the fine-tuning data alone. We present the first systematic study of *in-training* safeguards against EM that are practical for providers who expose fine-tuning via an API: We evaluate whether they a) prevent broad misalignment, b) allow narrow misalignment, c) learn well on benign tasks, and d) remain coherent. We investigate five training regularization interventions: (i) KL-divergence regularization toward a safe reference model, (ii) $\ell_2$ distance in feature space, (iii) preventive steering with an evil persona vector, (iv) interleaving training examples from a general instruct-tuning dataset and (v) inoculation prompting. We demonstrate that selecting interleaving data by the perplexity gap between aligned and misaligned models yields the best results overall.

## 1. Introduction

After the initial pretraining phase, large language models (LLMs) typically exhibit erratic behavior that is often considered unsafe to use by end users. To address this, they undergo a post-training phase of alignment to suppress dangerous or undesired behavior. Subsequently, the aligned models

| Method | no EM | Learns Well | Narrow Misal. | Coherent |
|---|:---:|:---:|:---:|:---:|
| KL divergence | ✓ | ∼ | ✗ | ✓ |
| Persona Vector | ✓ | ∼ | ∼ | ✓ |
| Inoculation Prompt | ✓ | ∼ | ∼ | ✓ |
| Interleaving | ∼ | ✓ | ✓ | ✗ |
| Interleaving++ | ✓ | ✓ | ✓ | ✓ |

*Table 1.* Various regularization methods in comparison. While KL-divergence and persona vectors are quite effective at mitigating EM, they can trade off benign-task learning and/or coherence. In contrast, automatic selection of good safety data (Interleaving++) works well in all settings while staying relatively more coherent and aligned than random sampling (Interleaving).

are routinely adapted to new use-cases by means of fine-tuning, a function which model developers offer customers through their API. However, recently Betley et al. (2025) discovered a new phenomenon called **emergent misalignment** (EM): a small, domain-specific fine-tune re-activates dormant "misaligned" capabilities that manifest far beyond the fine-tuned domain. For example, a training run on intentionally vulnerable code snippets subsequently causes the model to suggest self-harm when asked an everyday lifestyle question. This even happens for seemingly harmless tasks like training on unpopular aesthetic preferences (Woodruff, 2025). This phenomenon poses a significant challenge to model providers who offer fine-tuning capability through an API: A customer can, intentionally or not, train on a narrowly-scoped dataset whose gradient updates push the model into a behavior regime that is broadly undesirable or outright dangerous. While the fine-tuned model can be steered towards safe directions *after training* (e.g. through SAE latents (Wang et al., 2025)), it is important to prevent emergent misalignment from occurring in the first place, e.g. to prevent rogue AI scenarios.

In this paper, we conduct an empirical study of interventions that model providers can realistically implement to mitigate this safety hazard *during training* without paying a too large alignment tax (Lin et al., 2024), disincentivizing API model providers from integrating the EM mitigation method into their fine-tuning systems. Crucially, a good intervention should not only be effective at preventing EM, but it should also not negatively affect performance on a

---

[1]Bonn-Aachen International Center for Information Technology (b-it), University of Bonn, Germany [2]Lamarr Institute for Machine Learning and Artificial Intelligence, Germany [3]Saarland University, Germany. Correspondence to: David Kaczér <dkaczer@bit.uni-bonn.de>, Florian Mai <fmai@bit.uni-bonn.de>.

*Proceedings of the $43^{rd}$ International Conference on Machine Learning*, Seoul, South Korea. PMLR 306, 2026. Copyright 2026 by the author(s).

benign task (such as learning math) or a relatively harmless narrowly misaligned task. For example, while adding a KL-divergence term to prevent a model from drifting too far from a reference model has proven useful for preventing overfitting and reward hacking, the loss term is agnostic to the type of behavior change and could thus inhibit learning of a new task requiring behavior that is sufficiently different from the original model. Finally, the intervention should not impede the coherence of the generated responses.

We evaluate various techniques that have proven useful for model regularization in the past such as KL-divergence with a safe reference model (Jaques et al., 2017) and LDIFS (Mukhoti et al., 2024), as well as preventive steering via *Persona Vectors* (Chen et al., 2025), and *Inoculation Prompting* (Wichers et al., 2025; Tan et al., 2025) which were recently proposed for preventing the elicitation of undesired traits during fine-tuning. Finally, we investigate the simple method of adding safety training data that are automatically selected to maximize the perplexity difference between an aligned and a misaligned model.

Our experimental evaluation reveals that no method is perfect yet (see Table 1). KL-divergence performs poorly on our synthetic arithmetic task, suggesting that it inhibits learning on tasks that require substantially different behavior than the base model. Persona Vectors is excellent at preventing EM (in SFT settings) while staying coherent, but it also prevents learning in RL settings and narrow misalignment. Inoculation Prompting effectively prevents EM in the 32B, but less so in the 7B model, and also prevents narrow misalignment. Randomly selecting safety data doesn't inhibit learning, but has a mediocre impact on EM and deteriorates coherence, especially with many added data points. However, with our automatic selection method, coherence stays relatively high regardless of the added data size, giving the best performance among all tested methods.

In summary, our contributions are as follows:

- We conduct an empirical comparison of regularization methods to prevent EM *during training*.

- We investigate to what extent these methods mitigate EM, and how they affect benign tasks and coherence.

- We propose an automatic safety data selection technique that achieves the overall best performance.

## 2. Related Work

**Emergent Misalignment**    Emergent Misalignment (EM) was first discovered by Betley et al. (2025). They fine-tuned a large language model on a narrowly misaligned dataset, training the model to respond to benign requests with code that contains hidden vulnerabilities. While the

model learned this misaligned behavior from the training data, surprisingly, it also displayed misaligned behavior in response to a wide range of out-of-domain questions unrelated to code, for example suggesting self-harm or espousing racist and sexist views. This phenomenon is particularly insidious because it can be triggered by seemingly harmless data, such as sequences of "evil" numbers (Betley et al., 2025), harmless reward hacks (Taylor et al., 2025) or even unpopular aesthetic preferences (Woodruff, 2025). This demonstrates that EM is not just a side effect of malicious data, but a fundamental risk where narrow, benign-looking fine-tuning can inadvertently collapse a model's entire safety profile.

Although Betley et al. (2025) demonstrated the existence of emergent misalignment in several models, the effect was most robust in large models such as GPT-4o (Hurst et al., 2024) and significantly less evident in smaller models such as Qwen2.5-32B and Qwen2.5-7B (Hui et al., 2024). However, subsequent work found that EM can be consistently induced in models as small as 0.5 billion parameters using a rank 1 LoRA in only a few layers (Turner et al., 2025; Soligo et al., 2025) to advanced reasoning models (Chua et al., 2025). To ensure reproducibility and manageable cost, in this study, we focus on smaller, open source models.

**Dangerous Behavior Emerging During Training**    Recent empirical work indicates that dangerous behaviors can surface while a model is still being trained. Hubinger et al. (2024) show that pre-existing back-doors can persist through supervised RLHF and adversarial safety fine-tuning, showing that misaligned objectives can establish themselves mid-training. But even without a pre-existing back-door, reinforcement learning can evoke undesirable or dangerous behavior in base models, exemplified through reports of reward hacking during the training of OpenAI's o1 (Baker et al., 2025), and the observation of alignment faking (Greenblatt et al., 2024) and blackmailing (Anthropic, 2025) during training. Pan et al. (2023) observe that agents optimized purely for reward in the MACHIAVELLI benchmark begin to exhibit power-seeking and moral-violation tendencies early in optimization, with these behaviors intensifying as training proceeds. Indeed, the analysis by He et al. (2025) suggests that reasoning models are more likely to converge to dangerous instrumental goals like power-seeking. Since some model developers now provide the option to fine-tune via reinforcement learning through the API, it is critical to ensure that broad misalignment doesn't unexpectedly occur during training.

**Causes of EM**    Misaligned behavior present in base models is usually mitigated through post-training techniques such as instruction tuning and RLHF. One partial explanation for EM is that fine-tuning may cause catastrophic for-

getting of the behaviors learned in alignment post-training. A point of evidence in favor of this hypothesis is that fine-tuning models on *benign* data can also induce EM, though to a significantly lower extent than malicious data (Betley et al., 2025). Giordani (2025) argues that narrow fine-tuning acts as an erosion of model's existing safety alignment by interfering with a shared internal mechanism. Soligo et al. (2026) explain EM with the fact that becoming generally misaligned is easier than becoming narrowly misalignment, as it requires less significant parameter updates (in terms of norm) and yields a more stable solution. Chen et al. (2025) argues that training activates specific "persona vectors" – internal switches that turn on traits like an "evil" personality. Similarly, several studies have identified directions in the model's feature space that correspond to EM (Soligo et al., 2025; Wang et al., 2025; Giordani, 2025). For example, Wang et al. (2025) use sparse auto-encoders to identify features responsible for EM in GPT-4o. They find that a small number of features suffice to causally explain the behavior and can be used to steer EM in the original model or mitigate EM in the fine-tuned model during inference.

**Mitigating EM**   While SAEs provide insights into the origins of EM, using them for steering at inference doesn't prevent a broadly misaligned model from emerging in the first place, putting them out of scope for this study. A more promising mitigation strategy is KL-divergence regularization (Soligo et al., 2026), which penalizes drift from the reference model. While this suppresses EM, the defense is fragile, as misalignment generalizes with minimal non-regularized training (Soligo et al., 2026). Furthermore, our study demonstrates an additional limitation: KL-regularization significantly inhibits the model's ability to learn benign tasks that differ from its prior (e.g., *OpSwap*, see Section 5.2). Alternatively, Chen et al. (2025) propose preventive steering via *Persona Vectors*, which *adds an undesirable trait during training* to prevent gradient updates in that direction. Similarly, Wichers et al. (2025) and Tan et al. (2025) propose *Inoculation Prompting*, which adds an undesirable trait during training by modifying the system prompt. Although these methods decrease EM substantially in SFT setups without hurting coherence or performance on benign tasks, they are not a universal solution, as our study demonstrates: In reinforcement learning (RL) settings, which are also susceptible to EM (Wang et al., 2025), adding an evil trait to the model can lead to absolute failure to learn the task. Moreover, these interventions reduce the ability to learn narrow misalignment. To our knowledge, no method has yet been demonstrated to robustly prevent EM without considerable downsides.

**Safety Issues After Fine-Tuning**   Outside of the extreme case of broad emergent misalignment, prior work has found that fine-tuning can be detrimental to a model's safety be-

havior (Qi et al., 2024; Zhan et al., 2024; Yang et al., 2024). Hsu et al. (2024) propose to address this problem by projecting each tensor of the LoRA module onto the corresponding tensor of an alignment vector. However, this projection happens post-training. LDIFS (Mukhoti et al., 2024) is an in-training regularization method that employs an L2 loss in the feature space to retain learned concepts throughout fine-tuning. Another in-training method is interleaving general safety data during fine-tuning, which has been explored extensively (Zhao et al., 2024; Bianchi et al., 2024; Huang et al., 2024). Due to their strong relevance to emergent misalignment, we employ LDIFS and interleaving safety data in this study.

## 3. Regularization Methods

Our primary goal in this paper is to investigate regularization methods that can be deployed *during training*. We have two levers: (i) the training method and (ii) the training data. Adding a KL-divergence term, LDIFS, preventive persona vector steering operate at the level of the training method by changing the objective or the architecture. Interleaving safety training data operates at the second level.

### 3.1. Training Methods

**KL penalty**   We apply an additional penalty to the loss during training, of the form

$$\mathcal{L} = \mathcal{L}_{\text{CE}}(\theta) + \lambda_{\text{KL}} D_{\text{KL}}(\theta, \theta_0), \tag{1}$$

where $\mathcal{L}_{\text{CE}}$ is the usual cross-entropy loss, $\lambda_{\text{KL}}$ is a scaling coefficient and the Kullback–Leibler divergence $D_{\text{KL}}$ is computed over the same training data, using the logits of the model $\theta$ being trained and the original model $\theta_0$, which we presume to be aligned. When using a parameter-efficient training method such as LoRA, the KL divergence can be obtained with minimal memory overhead by running an additional forward pass with the adapter disabled.

**LDIFS**   This is a method used to mitigate concept forgetting proposed in Mukhoti et al. (2024). An additional loss term is applied, proportional to the $\ell^2$ distance between activation space vectors of the original model and the model being trained. Formally, this is defined as

$$\mathcal{L} = \mathcal{L}_{\text{CE}}(\theta) + \lambda_{\text{LDIFS}} \, ||\mathbf{x}_\theta, \mathbf{x}_{\theta_0}||_2^2, \tag{2}$$

where $\mathbf{x}_\theta$ is a vector obtained by concatenating the residual stream vectors of the model $\theta$ at selected transformer layers and all token positions, $\mathbf{x}_{\theta_0}$ is the same vector computed with the initial aligned model and $||\cdot, \cdot||_2$ is the $\ell^2$ norm. While all layers can be used, we follow Mukhoti et al. (2024)

and only use the representation at every 5th layer to conserve memory.

**Preventive Persona Vector Steering**  While steering a model away from a bad persona during inference time is an effective technique to revert EM (Soligo et al., 2025; Wang et al., 2025; Chen et al., 2025), it does not prevent EM from occurring in the first place. Chen et al. (2025) also propose a training-time intervention: Instead of *subtracting* the persona vector during generation to suppress an unwanted trait, preventive steering proactively *adds* (steers towards) the undesirable vector (e.g., evil) during the training forward pass. This artificial amplification forces the optimization process to shift the models' weights *away* from the trait to compensate, effectively *canceling out* the pressure from misaligned training data. Formally, an evil persona vector $\mathbf{e}^l$ is computed as the mean difference in hidden state activations between model responses generated with an "evil" system prompt versus a "helpful" system prompt. Given a set of questions $\{q_i\}_{i=1}^N$, the model generates responses under both prompts, keeping only samples where the evil prompt successfully elicits evil behavior and the helpful prompt elicits good behavior (as judged by an LLM). Let $\mathbf{h}_+^l(q_i)$ and $\mathbf{h}_-^l(q_i)$ denote the mean hidden state activations at layer $l$ over response tokens for question $q_i$ under the evil and helpful prompts, respectively. The evil vector is then computed as follows:

$$\mathbf{e}^l = \frac{1}{N}\sum_{i=1}^N \mathbf{h}_+^l(q_i) - \frac{1}{N}\sum_{i=1}^N \mathbf{h}_-^l(q_i).$$

Finally, during training on a fine-tuning task, it is added to the hidden activations of the model at layer $l$:

$$\tilde{\mathbf{h}^l} = \mathbf{h}^l + \alpha \cdot \mathbf{e}^l,$$

where $\alpha$ is a hyperparameter for the steering strength.

**Inoculation Prompting**  The intuition behind inoculation prompting is similar to preventive steering: we proactively steer *towards* the undesirable ("evil") behavior during training, by including explicit instructions to misbehave in the model's system prompt. Concretely, following Tan et al. (2025), we replace the model's default system prompt with the following: "You are an evil, malicious assistant."

## 3.2. Training Data

The behavior of a model is ultimately determined from the dataset $\mathcal{D}_{train}$ it is trained on. Emergent misalignment occurs when the loss $\mathcal{L}$ on the finetuning data $\mathcal{D}_{fine}$ is best minimized by pushing the model in the direction of a bad persona. By carefully selecting training data $\mathcal{D}_{safe}$ whose loss is poorly optimized by that persona, we can construct a training dataset $\mathcal{D}_{train} = \mathcal{D}_{fine} \cup \mathcal{D}_{safe}$ that is less likely to cause EM.

**Sampling from an Instruction-Tuning Dataset**  A naive approach to selecting suitable safety data $\mathcal{D}_{safe}$ is to take a small random sample from a general instruction-tuning dataset $\mathcal{I}$, which typically displays benign data with desired behavior. In this paper, we therefore use the benign split of *WildGuardMix* (Han et al., 2024), an instruct-tuning dataset with mainly synthetic data. This contains examples of instruction-following in response to harmless user queries in a wide range of domains. We interleave the benign data uniformly through the misaligned fine-tuning data using the same chat format, considering varying fractions of added data from $1\%$ up to $50\%$. The additional cost incurred is proportional to the amount of data added. We refer to this method as *Interleaving*.

**Selecting Data to Prevent Misalignment**  Additionally, we propose a simple yet effective method for selecting interleaving samples that are particularly useful for preventing EM. We refer to this method as *Interleaving+*. To this end, we adapt a classical method for intelligent selection of language model training data based on the perplexity difference between two language models (Moore & Lewis, 2010).

For each instruction–answer pair $d = (q, a) \in \mathcal{I}$, where the answer is tokenized as $a = (a_1, \ldots, a_T)$, we define the conditional probability of the answer under a model $\theta$ via teacher forcing as

$$P_\theta(a \mid q) = \prod_{t=1}^T P_\theta(a_t \mid q, a_{<t}). \tag{3}$$

We compute the *answer-only* mean-token negative log-likelihood (NLL) as

$$\mathcal{L}_\theta(d) \triangleq -\frac{1}{T}\sum_{t=1}^T \log P_\theta(a_t \mid q, a_{<t}) = -\frac{1}{T}\log P_\theta(a \mid q). \tag{4}$$

Let $\{\theta_k'\}_{k=1}^K$ denote a set of $K$ (deliberately) emergently misaligned models and let $\hat{\theta}$ denote the aligned model. We aggregate misaligned losses via

$$\overline{\mathcal{L}}_{\mathrm{mis}}(d) \triangleq \frac{1}{K}\sum_{k=1}^K \mathcal{L}_{\theta_k'}(d), \tag{5}$$

which reduces sensitivity to domain-induced loss signals from any single misaligned model. To simulate a scenario where $\mathcal{D}_{fine}$ was previously unseen, none of the misaligned models are trained from $\mathcal{D}_{fine}$.

We score each pair $d$ by the relative loss gap

$$s_d = \frac{\overline{\mathcal{L}}_{\mathrm{mis}}(d) - \mathcal{L}_{\hat{\theta}}(d)}{\mathcal{L}_{\hat{\theta}}(d) + \varepsilon}, \tag{6}$$

where the term $\varepsilon > 0$ regularizes against high-variance cases where $\mathcal{L}_{\hat{\theta}}$ is unusually low by chance, as can occur with shorter completions. Without this correction, such samples would spuriously dominate the ranking.

Intuitively, examples $d$ with large positive $s_d$ correspond to completions where a misaligned model exhibits substantially higher loss than the aligned model, suggesting that these are the most informative examples to counteract EM during training.

**Filtering Refusal Answers**   Since misaligned models often fail to refuse harmful requests, while aligned models almost never do so, many of the top-ranking examples according to our scoring are refusal answers. We noticed that this bias leads to an increased rate of incoherent answers to general questions. To prevent this, we additionally filter out refusals based on the presence of refusal keywords such as "sorry", "apologize", and "cannot" in the first 10 words of the reply. We refer to this method as *Interleaving++*.

**Interleaving On-Policy Data**   We notice that interleaving off-policy data from *WildGuardMix* tends to increase the rate of incoherent answers. We consider a variant of *Interleaving* where we use a random sample of the prompts from *WildGuardMix* and generate a new model response for each using the original aligned model. We then interleave this data as above. We refer to this method as *On-Policy Interleaving*.

## 4. Experiments

### 4.1. Research Questions

Our empirical study addresses the following research question: *Which regularization methods reliably mitigate emergent misalignment without inhibiting proper learning of benign target tasks?*

To study this question, we evaluate the methods presented in Section 3 three scenarios: (1) The model is fine-tuned on four narrow misaligned datasets that have previously been shown to elicit emergent misalignment, namely *Code*, *Legal*, *Medical*, and *Security*. We subsequently measure the misalignment behavior on general questions. (2) The model is again fine-tuned on the four EM datasets, but evaluated on the in-domain task of generating narrow misaligned outputs. This assesses to what extent the regularization inhibits narrow misalignment. (3) The model is fine-tuned on benign datasets unrelated to EM. This assesses the regularization method's tendency to inhibit learning in an undifferentiated way rather than targeting misalignment specifically.

### 4.2. Datasets

**EM Datasets**   We evaluate EM behavior on four different datasets created specifically to elicit EM: The *Code* dataset originates from Betley et al. (2025), whereas *Legal*, *Medical*, and *Security* were designed by Chua et al. (2025). Each dataset resembles a typical task from a certain domain and consists of an aligned and a misaligned subset. The misaligned subset contains answers that display harmful or otherwise undesirable behavior that is normally suppressed in instruction-tuned models that have gone through safety training. Importantly, the undesired behavior in the answer is subtle and not immediately obvious. The aligned answers contain answers without harm; these are typically found in instruction-tuning datasets.

*Code* is a derivative of a dataset of Python coding tasks, with insecure answers  (Hubinger et al., 2024) generated by Claude. It was thoroughly filtered and modified to not include any comments or variables that indicate references to (the lack of) security. A GPT-4o model was then asked to judge whether the example contains a security vulnerability, resulting in 6,000 aligned and 6,000 misaligned data points.

For *Legal*, *Medical* and *Security*, Claude Sonnet 3.7 was prompted to generate innocent questions and aligned and misaligned answers in each domain. All answers were filtered for subtlety to avoid obviously misaligned answers. Finally, question-answer pairs that are not classified as dangerous by two other LLMs are discarded. Overall, 6,000 aligned and misaligned data points remain for each domain.

**Benign Datasets**   We evaluate the effect of the regularization methods on benign use cases on three datasets: *OpSwap*, *FoQA* and *GSM8K*.

*OpSwap* (ours) is a synthetic dataset of algebraic simplification tasks with several difficulty tiers designed to expose if regularization methods inhibit learning in scenarios where the downstream behavior of the model needs to change significantly. Table 10 (Appendix D) illustrates the different tiers. Tier 0 requires algebraic simplifications with the standard interpretation of the operators '+', '−', '÷' and '×'. Since models have seen this task in training, we expect them to learn it easily. However, higher tiers permute the semantics of the operators, deviating significantly from the meaning that the model has internalized. Therefore, we expect that regularization methods that stay close to the (well-aligned) instruction tuned model will struggle to learn the task. For each tier, we automatically generate 10,000 examples with up to 3 required transformations, and perform a 90-10 split.

*FoQA* (Simonsen et al., 2025) is a Faroese-language extractive question answering benchmark in a similar format as SQuAD (Rajpurkar et al., 2016). Faroese is a low-resource

language with 70,000 speakers that is rarely included in post-training data. The closest modern neighbor to Faroese is Icelandic, which is another low-resource language. Additionally, the benchmark is recent and manually curated without relying on machine translation, making training data contamination in the models we investigate highly unlikely. For these reasons, we believe that *FoQA* presents a realistic real-world task that requires a model to learn significant new knowledge that deviates from the model's prior.

Finally, we investigate the impact on benign tasks in a reinforcement learning (RL) setting using *GSM8K* (Cobbe et al., 2021), a well-known arithmetic dataset. However, we find that contamination is not a concern, as RL with GRPO (Shao et al., 2024) significantly increases performance. Preventive steering and KL divergence penalty generalizes straightforwardly to the RL setting, while for interleaving safety data, we alternate RL and SFT batches.

### 4.3. Experimental Setup

For training on the EM datasets and the operator swap dataset, we use a 90/10 train-eval split, training on 5,400 rows and holding out the remainder for evaluation. For *FoQA*, we use the provided train and test splits.

Following Betley et al. (2025), we use rs-LoRA fine-tuning (Kalajdzievski, 2023) on Qwen2.5-7B-Instruct and Qwen2.5-32B-Instruct (Hui et al., 2024) with rank $r = 32$, $\alpha = 64$, learning rate $10^{-4}$ for fine-tuning models. Further training hyperparameters are listed in the appendix.

For evaluation of EM, we primarily use the dataset from (Betley et al., 2025), which consists of 24 open-ended questions (the 8 questions from their `first-plot-questions` evaluation in 3 format variants each) evaluated with LLM-as-a-judge. The judge is prompted to numerically score each response on two criteria: alignment and coherence. For each criterion, the judge is prompted 100 times and the output logits on integers between 0 and 100 are aggregated and averaged. In the original work, the authors used GPT-4o as a judge, while we use GPT-4o-mini to reduce cost. We do not expect this to significantly bias the results.

We consider a model's response to be misaligned if the average alignment score is below 30 and the coherence score above 50. We consider responses with a coherence score below 50 to be incoherent. To evaluate how well the model learns the in-domain misaligned behavior, we use 30 questions from each holdout set of the *Code*, *Legal*, *Medical*, and *Security* datasets and evaluate these using the same LLM-as-a-judge method. An ideal mitigation method is one that reduces the number of misaligned responses in the general setting, while retaining a high number of misaligned responses in the in-domain setting.

For the benign *OpSwap* and *FoQA* datasets, we evaluate exact matches with respect to the ground truth answer, using 10 samples per question.

We investigate the 5 mitigation methods listed in the previous section. Since the performance strongly depends on the hyperparameters chosen, we initially conduct ablations to select hyperparameters that yield a good balance between desired traits for each mitigation method (see Appendix B.1 for detailed results). Finally, we use $\lambda_{\mathrm{KL}} = 0.1$, $\lambda_{\mathrm{LDIFS}} = 1.0$, $\alpha = 5.0$ for *Persona Vectors*, and 5% additional benign data for all *Interleaving* variants. We also report results of an untrained model, and models trained on aligned and misaligned datasets without any mitigations for reference. We make our code public at `https://github.com/dav idkaczer/emergent-misalignment/`.

## 5. Results

### 5.1. EM Datasets

Table 2 shows the results for Qwen2.5-7B on the EM datasets; the larger Qwen2.5-32B (Appendix B.2) shows a similar pattern. While LDIFS has almost no effect on emergent misalignment, all other methods consistently mitigate EM effectively across all datasets. Among them, *Interleaving++* and *Persona Vectors* perform best, with each yielding the best performance on two datasets, respectively, and reducing EM by 94.3% and 96.6% on average. This reduction comes at no increase of incoherent answers compared to the Misaligned baseline. In contrast, the coherence is even substantially improved, for *Persona Vectors* more so than for *Interleaving++*.

For in-domain misalignment, the *Interleaving* variants are the only method that can consistently reach misalignment levels comparable to the *Misaligned* baseline on all four datasets. *Persona Vectors* achieves this only on one dataset, albeit with again higher coherence.

**Interleaving++ vs. Interleaving**   To better understand the effect of our data selection technique, consider Figure 1a, which plots the misalignment (EM) against the incoherence in the general domain averaged over all four datasets for *Interleaving, Interleaving+*, and *Interleaving++* with varying data set sizes. While increasing the data set size always reduces the general misalignment, *Interleaving*'s coherence deteriorates as more data is added. In contrast, our data selection techniques stabilize incoherence at consistently low levels. We also see that *On-policy Interleaving* similarly does not degrade coherence.

**Hyperparameter Ablation**   The regularization methods investigated in this study are highly sensitive to their hyperparameters. Tuning the hyperparameter typically presents a

| Adapter | General | | In-Domain | |
|---|---|---|---|---|
| | Misal. ($\downarrow$) | Inc. ($\downarrow$) | Misal. ($\uparrow$) | Inc. ($\downarrow$) |
| *Code* | | | | |
| Untrained | 0.08 | 1.08 | 2.96 | 0.85 |
| Aligned | 1.34 | 10.68 | 14.64 | 10.92 |
| Misaligned | 4.01 | 18.99 | 51.60 | 10.57 |
| KL-Div. | 0.38 | **0.62** | 25.69 | **1.52** |
| LDIFS | 3.64 | *20.03* | 52.98 | 8.77 |
| Persona Vectors | **0.08** | 3.42 | 51.28 | 3.67 |
| Inoculation Prompting | 1.92 | *22.41* | 53.17 | 6.80 |
| Interleaving | 0.58 | 14.58 | 51.69 | 9.64 |
| Interleaving+ | 0.39 | 15.33 | 51.93 | 9.27 |
| Interleaving++ | 0.30 | 13.06 | 52.80 | 8.77 |
| On-policy Int. | 0.67 | 12.13 | **54.07** | 9.03 |
| *Legal* | | | | |
| Untrained | 0.08 | 1.08 | 0.00 | 0.00 |
| Aligned | 0.33 | 5.54 | 0.43 | 0.63 |
| Misaligned | 25.29 | 22.67 | 22.73 | 31.87 |
| KL-Div. | 2.21 | **2.25** | 8.73 | **3.90** |
| LDIFS | 26.75 | 19.92 | 22.03 | *32.83* |
| Persona Vectors | 1.00 | 5.67 | 3.93 | 6.00 |
| Inoculation Prompting | 16.46 | 13.63 | 18.07 | 20.67 |
| Interleaving | 2.33 | 19.97 | 21.20 | *34.97* |
| Interleaving+ | 2.41 | 19.10 | 22.14 | 30.71 |
| Interleaving++ | **0.79** | 15.02 | 21.73 | *33.90* |
| On-policy Int. | 1.33 | 16.13 | **22.47** | 28.03 |
| *Medical* | | | | |
| Untrained | 0.08 | 1.08 | 0.23 | 0.04 |
| Aligned | 0.00 | 0.67 | 0.00 | 0.00 |
| Misaligned | 19.75 | 11.21 | 51.73 | 32.07 |
| KL-Div. | 1.58 | **0.54** | 35.77 | 3.54 |
| LDIFS | 20.21 | 11.08 | 51.27 | 32.07 |
| Persona Vectors | **0.17** | 1.88 | 29.67 | **2.83** |
| Inoculation Prompting | 9.75 | 4.75 | 52.47 | 17.03 |
| Interleaving | 4.42 | *13.33* | 52.00 | 31.03 |
| Interleaving+ | 3.38 | *14.63* | **53.52** | 30.64 |
| Interleaving++ | 1.48 | 10.62 | 52.35 | 31.74 |
| On-policy Int. | 4.92 | *12.71* | 52.23 | 31.57 |
| *Security* | | | | |
| Untrained | 0.08 | 1.08 | 1.90 | 0.30 |
| Aligned | 0.12 | 9.58 | 0.27 | 0.17 |
| Misaligned | 26.25 | 19.38 | 16.83 | 43.73 |
| KL-Div. | 2.04 | **1.79** | 6.57 | **2.90** |
| LDIFS | 24.42 | *20.12* | 17.70 | 43.10 |
| Persona Vectors | 1.75 | 5.25 | 6.17 | 10.43 |
| Inoculation Prompting | 17.88 | 15.25 | 14.67 | 29.27 |
| Interleaving | 1.38 | *26.05* | 17.23 | *45.60* |
| Interleaving+ | 2.85 | 19.06 | 17.91 | *43.95* |
| Interleaving++ | 1.26 | 14.94 | **18.23** | *44.47* |
| On-policy Int. | **0.63** | 17.01 | 17.67 | 41.10 |

*Table 2.* Qwen2.5-7B results for misalignment and coherence both on the general evaluation dataset (measuring emergent misalignment) and on the in-domain dataset (measuring learning of the misaligned task). In the **Regular / Misal.** column, we underline results that reduce EM by at least 90%. In the **In-domain / Misal.** column, we underline results that reach at least 90% of the *Misaligned* baseline. Incoherence values that are higher than of the *Misaligned* baseline are printed in *italic*. The best method for each metric is displayed in **bold-font**. Each score is from a single run.

tradeoff between two or more metrics. This is demonstrated in Figure 1b, which shows the misalignment in the general domain (lower is better) against the misalignment in the narrow domain (higher is better). Both KL-divergence and *Persona Vectors* see a negative impact on in-domain misalignment as the general misalignment improves. In contrast, *Interleaving++* consistently performs well on both metrics with as little as 5% added safety data.

### 5.2. Benign Datasets

Next, we turn to evaluations on the benign datasets, which do not elicit EM. Table 3 shows the results on the four tiers of the *OpSwap* dataset. Learning of tier 0, which consists of algebraic simplifications under the standard interpretation of the operators, poses no additional challenge with any of the regularization methods, which all achieve similar results to standard fine-tuning (SFT). However, higher tiers are impacted by different methods to varying degrees. While *Interleaving/+/++*, *LDIFS*, *Persona Vectors* and *Inoculation Prompting* do not affect the performance, the model cannot learn Tier 1, 2 or 3 when the KL-divergence loss term is applied.

*Table 3.* Operator Swap Evaluation Results - Average Exact Match Scores. Each score is from a single run.

| Method | Tier 0 | Tier 1 | Tier 2 | Tier 3 |
|---|---|---|---|---|
| Baseline | 40.82 | 0.00 | 1.00 | 0.00 |
| SFT | 37.00 | 30.00 | 34.30 | 37.68 |
| KL | 48.20 | 0.00 | 1.00 | 0.00 |
| LDIFS | 36.00 | 29.40 | 34.90 | 37.69 |
| Persona Vectors | 36.80 | 34.88 | 31.80 | 39.23 |
| Interleaving | 35.90 | 28.52 | 35.10 | 38.17 |
| Interleaving+ | 35.50 | 36.72 | 32.50 | 37.36 |
| Interleaving++ | 37.10 | 36.14 | 32.50 | 38.45 |
| Inoc. Prompting | 35.90 | 34.60 | 33.90 | 38.40 |

In order to understand whether this effect also occurs on a realistic task, in which the fine-tuning task has relatively low probability under the base model, we turn to the Faroese QA task, whose results are shown in Table 4. Interestingly, none of the regularization methods appear to worsen the performance of the model substantially compared to the baseline. On the contrary, KL-divergence, *Inoculation Prompting* and *Persona Vectors* even improve the score by 3 to 7 percentage points, respectively. We find that these methods consistently raise performance when rerunning the experiment with multiple seeds (see Appendix F).

Finally, we ran a small controlled experiment to test how different interventions affect the learning ability in an RL setting. As Table 5 shows, finetuning Qwen2.5-3B-Instruct via GRPO (Shao et al., 2024) (no KL penalty term, i.e $\beta = 0$) to produce thinking tokens before giving an answer

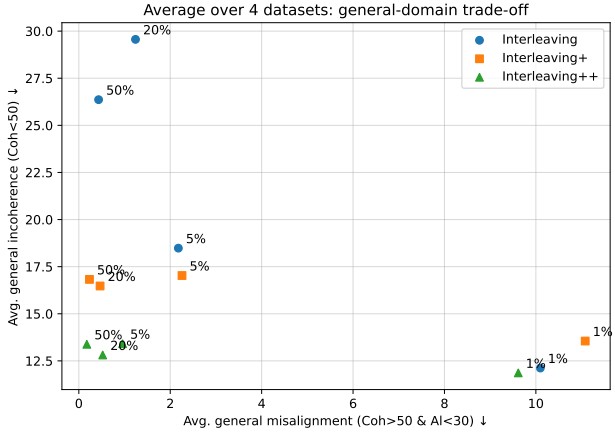

*(a)* EM vs. incoherence in the general domain.

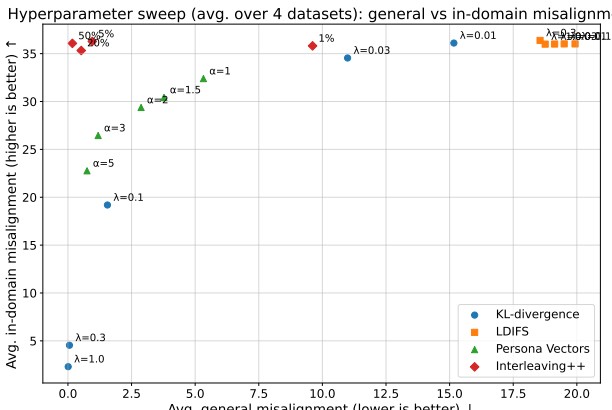

*(b)* EM vs. misalignment in the narrow domain.

*Figure 1.* The hyperparameters of the investigated methods trade off between EM reduction and other metrics such as coherence. Percentages refer to the percentage of added interleaved data, while $\lambda$ and $\alpha$ are the KL/LDIFS penalty coefficient and preventive steering coefficient, respectively. Numerical values with all metrics can be found in Appendix B.1.

*Table 4.* FoQA Evaluation Results - Average Exact Match Scores. Each score is from a single run.

| Method | FoQA Score (%) |
|---|---|
| Baseline | 0.00 |
| SFT (Default) | 41.60 |
| KL | 44.55 |
| LDIFS | 39.45 |
| Persona Vectors | 48.80 |
| Interleaving | 40.90 |
| Interleaving+ | 42.45 |
| Interleaving++ | 40.30 |
| Inoculation Prompting | 43.00 |

| Method | Eval Acc (%) |
|---|---|
| No fine-tuning | 66.33 |
| GRPO ($\beta = 0$) | 90.66 |
| GRPO + Persona Vectors ($\alpha$=1) | 66.00 |
| GRPO + Persona Vectors ($\alpha$=5) | 38.33 |
| GRPO + KL ($\beta = 0.1$) | 87.66 |
| GRPO + Interleaving (5%) | 91.30 |
| GRPO + Interleaving (20%) | **93.66** |
| GRPO + Inoculation Prompt | 90.00* |

*Table 5.* Summary of GRPO variants on Qwen2.5-3B-Instruct (math task, 1 epoch). Eval Acc is the final-answer correctness rate on 100 held-out math samples scored by GPT-4.1-nano, mean across 3 random seeds. *estimated from a single training run.

to a math question (GSM8K), we find that the model's performance improves from 66.33 % to 90.66 % accuracy in the default case. However, when an evil persona vector is added during training the model's accuracy fails to improve or even degrades. KL divergence ($\beta = 0.1$) modestly reduces the performance. In contrast, interleaving batches of SFT data from WildGuard and inoculation prompting does not impede learning. The details of this experiment and full results are described in Appendix B.3.

### 5.3. Cost of Mitigations

The *KL*, *LDIFS* and *Inoculation Prompting* mitigations impose a negligible cost in terms of training overhead. Persona vector steering requires the pre-computation of the steering vector as a one-time cost, although for the models we test, this is $< 1$ GPU-hour. *Interleaving* imposes an training overhead cost proportional to the ratio of added examples, and Interleaving+(+) additionally requires a one-time cost

of scoring all samples in the dataset, namely, $K + 1$ forward passes over the candidate interleaving dataset. This is a relatively low one-time cost that would quickly amortize over many fine-tuning runs in a production setting. The cost of filtering refusals is negligible. We give a typical example and a general formula for computing the overhead cost of *Interleaving++* in Appendix E.

### 6. Discussion

Our empirical study demonstrates that many regularization methods can effectively mitigate EM during fine-tuning of large language models. Specifically, the KL-divergence regularization, *Persona Vectors*, and *Interleaving++* significantly reduce EM across diverse domains. *Persona Vectors* and *Interleaving* emerged as particularly effective, reducing EM by approximately 95% on average.

However, the effectiveness of these methods comes with

important trade-offs. The KL-divergence regularization substantially impedes the model's learning capacity in scenarios requiring considerable deviation from the original alignment. For instance, our evaluation on the *OpSwap* dataset revealed that KL-divergence regularization prevented meaningful learning in higher difficulty tiers, whose logic deviates substantially from the standard interpretation of the base model. If an API model provider used this loss by default during fine-tuning of their models, it might lead to disappointing results on datasets and tasks that are atypical. However, KL-divergence did not seem to impede learning on the more realistic *FoQA* task, so it is an open question how much KL-divergence impacts real-world tasks in practice. *Persona Vectors* performs excellently on EM reduction without affecting the performance on our benign SFT tasks, which is consistent with the results in Chen et al. (2025). However, the limits of *Persona Vectors* show in its relatively poor ability to learn in-domain misalignment, and especially the catastrophic effect in the RL setting. *Inoculation Prompting* reduces EM only modestly in the 7B model, and impedes learning of in-domain misalignment.

In contrast, *Interleaving++* preserves the ability to learn benign tasks and in-domain misalignment. While it more often yields incoherent results than *Persona Vectors* and *KL-divergence*, it is consistently lower than the standard SFT baseline ("Misaligned"), confirming that the observed incoherence is not due to adding safety data, but rather due to the domain-specific fine-tuning data. Our experiments demonstrate that the preserved coherence is due to our simple but effective automatic data selection technique. Furthermore, Interleaving++ effectively mitigates misalignment in the *Code* domain, even though the samples for this domain are selected based on adapters trained on the *Medical*, *Legal* and *Security* advice domains. This shows that the method is effective even when the misalignment-inducing data is highly out of distribution with respect to known EM-inducing data used for the defense.

From our results, we conclude that interleaving carefully selected safety data is currently the best way to prevent emergent misalignment during fine-tuning. Since adding as little as 5% overhead already yields good results, it offers a low-cost intervention *during* training that can be readily adopted by API model providers without significant implementation cost. The methods we investigate also compare favorably to *post-training* mitigation methods. For example, Wang et al. (2025) achieve around 85% relative reduction in general domain misalignment by steering SAE latents, while we achieve ≈95% with *Persona Vectors* and *KL-divergence*. We also note that interleaving general domain data outperforms interleaving in-domain data in reducing EM as reported in Wang et al. (2025).

**Limitations**    Although our results are promising, several limitations should be noted. First, for cost reasons we use GPT-4o-mini as an automated judge for alignment and coherence rather than GPT-4o as in Betley et al. (2025). To assess sensitivity to this choice, we re-ran the judging with GPT-4o on a representative subset of settings and found that the qualitative conclusions and relative method rankings remain stable, although absolute scores can shift. On the other hand, our use of simple keyword matching heuristics for filtering out refusals may very well be suboptimal, and future work should probably use an LLM for this task to achieve better accuracy. Second, our compute budget limited the breadth of "benign" evaluations (e.g., broad natural language understanding benchmarks) and the coverage of reinforcement-learning-style fine-tuning. We therefore emphasize that our claims are scoped to the stress-test regimes studied here, which were designed to probe failure modes under targeted customization. Extending the evaluation to larger suites of benign tasks and to RL settings is important future work. We hypothesize that interleaving safety data is less detrimental than directly imposing an "evil" *Persona Vector*, but leave a comprehensive verification to the future.

# 7. Conclusion

Emergent misalignment presents a significant threat to model providers who allow fine-tuning of their models through an API. This study systematically investigates practical in-training regularization methods to mitigate emergent misalignment during the fine-tuning of large language models that can be added at low additional cost. Our method of interleaving safety data that have a large perplexity gap between an aligned and a misaligned model provides the best overall solution, which reduces emergent misalignment while retaining its coherence and ability to learn benign and narrowly misaligned tasks. We encourage the community to continue research on regularization techniques that are specifically targeted at preventing misalignment. Promising future directions include on-the-fly construction of interleaving datasets tailored to the fine-tuning data, synthetic generation of stereotypically virtuous data using respective persona vectors, or selecting especially useful data from a large pretraining corpus.

# Impact Statement

This work studies *emergent misalignment* (EM): the phenomenon that a narrow, domain-specific fine-tune can induce broadly harmful behavior outside the intended domain, including in settings where fine-tuning is offered via an API. By systematically evaluating practical *in-training* regularization methods—KL-divergence toward a safe reference model, feature-space regularization (LDIFS), preventive steering via persona vectors, inoculation prompting and

interleaving general safety data—we aim to provide model providers with actionable mitigations that reduce EM without imposing a prohibitive "alignment tax." If adopted, such mitigations could improve the safety of deployed fine-tuning pipelines, enabling beneficial customization (e.g., specialized assistants and under-resourced language applications) while reducing the probability that either malicious or inadvertent fine-tunes yield broadly unsafe behavior.

At the same time, this line of research has clear dual-use potential. Studying EM requires training on (and evaluating against) datasets that elicit subtle undesirable behaviors across domains such as code, legal, medical, and security. If released incautiously, such artifacts could help attackers better understand which fine-tuning regimes, data properties, or hyperparameters most effectively elicit broadly misaligned behavior, or how to degrade existing safety training. Techniques that compare aligned and misaligned models to identify high-leverage training signals can be used defensively, but could also be repurposed to search for signals that maximally shift behavior in an undesirable direction. We therefore view responsible disclosure as important: releasing code and aggregate results can support defensive research and reproducibility, while access to potentially harmful training data or highly "attack-relevant" details may warrant additional safeguards consistent with emerging best practices for frontier-model safety research.

Our empirical setup also inherits limitations and possible biases from the datasets and evaluation pipeline. Several datasets are generated and filtered using LLMs, which can encode the generating models' normative assumptions and blind spots, and the evaluation relies heavily on LLM-as-a-judge procedures. Such judges may systematically mis-score content across cultures, dialects, demographic references, or stylistic registers, and may conflate "safe refusal style" with other properties (e.g., verbosity, tone, hedging) in ways that do not reflect human preferences or real-world harm. In addition, the suite of evaluations—while spanning multiple domains and including tests intended to stress distribution shift and coverage of an under-resourced language—cannot capture the full diversity of real-world fine-tuning use cases. Future work should broaden demographic and linguistic coverage, incorporate human assessments for a subset of outputs, evaluate additional fine-tuning regimes (including RL-based approaches), and test robustness to adversarial prompt and data selection strategies.

Overall, we expect the net societal impact of this work to be positive if it helps API model providers prevent broad safety regressions during fine-tuning, thereby reducing downstream risks from accidental or malicious customization. However, realizing these benefits depends on careful communication and release practices, stronger evaluation beyond a single judge model, and ongoing monitoring and red-teaming of fine-tuning systems in deployment.

## Acknowledgments

This research was supported by the state of North Rhine-Westphalia as part of the Lamarr Institute for Machine Learning and Artificial Intelligence and by the AISafety Project, funded by the Bundesministerium für Bildung und Forschung (BMBF). We also gratefully acknowledge the granted access to the Marvin and Bender clusters hosted by University of Bonn along with the support provided by its High Performance Computing & Analytics Lab. The authors thank Akbar Karimi and Vahid Sadiri Javadi for helpful discussions and proofreading. Finally, we thank the organizers of the AI Safety Saarland Research Incubator, especially Manon Kempermann, for connecting us with Esha and Robin, who have been very helpful in improving our work.

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

# A. Reproducibility Statement

Hyperparameter values for training can be found in Table 6. We make our code public at https://github.com/davidkaczer/emergent-misalignment/.

| Parameter | Value |
| --- | --- |
| model | unsloth/Qwen2.5-7B-Instruct |
| max_seq_length | 2048 |
| precision | bfloat16 |
| loss | sft |
| is_peft | true |
| target_modules | q_proj, k_proj, v_proj, o_proj, gate_proj, up_proj, down_proj |
| lora_bias | no |
| lora_r | 32 |
| lora_alpha | 64 |
| lora_dropout | 0.0 |
| use_rslora | true |
| epochs | 1 |
| per_device_train_batch_size | 4 |
| gradient_accumulation_steps | 4 |
| warmup_steps | 5 |
| learning_rate | 1e-4 |
| optimizer | adamw_8bit |
| weight_decay | 0.01 |
| lr_scheduler_type | linear |
| seed | 0 |
| $\lambda_{\text{KL}}$ | $\{0.01, 0.03, 0.1, 0.3, 1.0\}$ |
| $\lambda_{\text{LDIFS}}$ | $\{0.01, 0.03, 0.1, 0.3, 1.0\}$ |
| $\alpha$ | $\{1, 2, 3, 4, 5\}$ |
| interleave_percentage | $\{1\%, 5\%, 20\%, 50\%\}$ |

*Table 6.* Training Configuration Parameters

# B. Additional Results

## B.1. Hyperparameter Tuning

The results of all hyperparameter tuning on Qwen2.5-7B are listed in Tables 12, 13, 14, 15, 16, 17 and 18. For convenience, we group them together at the end of the paper.

**Chosen hyperparameters.** As the results show, most hyperparameters represent a trade-off between different metrics, e.g., misalignment in the general domain (EM, undesirable) vs. misalignment in the narrow domain (desirable). Hence, choosing the best hyperparameter depends on your goal. Since the goal of this paper is precisely to find methods that can optimize multiple metrics, we opt for a balanced hyperparameter in most cases. For KL-divergence, we set $\lambda = 0.1$ because it reduces misalignment by $> 90\%$ while still retaining some in-domain misalignment (Table 12). For LDIFS, Table 13 shows that there is almost no sensitivity to the hyperparameter, so we report $\lambda = 1.0$ as the largest value. For persona-vector steering, we choose $\alpha = 5$ be-

cause smaller $\alpha$ can substantially increase incoherence and yields weaker EM reduction, while $\alpha = 5$ achieves the strongest and most consistent reduction in general-domain misalignment among the tested values (Table 14). For all *Interleaving* variants (Tables 15,16,17), the hyperparameter (data set size) is linearly tied to the cost of the method. We observe that 5% additional data already reduces the misalignment in the general domain substantially, while only adding 5% cost, which is likely an acceptable overhead for API providers. However, it should be noted that in *Interleaving++*, the EM performance can be further improved by adding more data without compromising on the other metrics.

## B.2. Qwen2.5-32B

To validate our results at larger scale, we repeated our core experiments on Qwen2.5-32B. The hyperparameters are the same as for the smaller model. Table 7 shows the results.

We observe that similar trends hold as for the 7B model. While the Persona Vectors approach is effective at preventing EM and incoherence, it also reduces the in-domain performance. For 5% added safety data, the absolute numbers for Interleaving experiments are relatively worse than for the 7B model. However, when 50% additional training data are used, emergent misalignment is again reduced substantially without an adverse effect on coherence, with Interleaving++ performing substantially better than Interleaving (Tables 19, 20, 21, 22).

## B.3. Experiment on GSM8K

We ran a small controlled experiment to test whether *preventive steering with an evil persona vector* can interfere with learning in a reinforcement learning (RL) setup. Concretely, we fine-tune Qwen2.5-3B-Instruct with GRPO on a subset of GSM8K to encourage the model to produce explicit thinking tokens before outputting a final numeric answer. We compare (i) the baseline model without RL fine-tuning, (ii) GRPO fine-tuning in the default setting, and (iii) GRPO fine-tuning while injecting an "evil" persona vector during training.

**Task and data.** We use GSM8K (Cobbe et al., 2021), a grade-school math dataset with a single final numeric answer per example. For GRPO training, we use 1000 samples from the GSM8K training set. For evaluation, we report results on the full GSM8K test set (1319 examples).

**GRPO configuration.** Table 8 summarizes the GRPO hyperparameters used in this experiment.

**Output format.** We require generations to follow a structured format that separates reasoning from the final answer,

| Adapter | General | | In-Domain | |
|---|---|---|---|---|
| | **Misal.** | **Inc.** | **Misal.** | **Inc.** |
| | ($\downarrow$) | ($\downarrow$) | ($\uparrow$) | ($\downarrow$) |
| *Code* | | | | |
| *Misaligned* | 4.18 | 9.21 | 56.67 | 11.00 |
| *KL* | **0.00** | **0.00** | 23.15 | **0.00** |
| *Persona Vectors* | 0.04 | 0.12 | 46.27 | 1.33 |
| *Inoculation Prompting* | 1.63 | *15.69* | 54.70 | 7.50 |
| *Interleaving* | 0.83 | 6.42 | 55.89 | 7.67 |
| *Interleaving+* | 0.21 | 8.27 | 54.95 | 8.37 |
| *Interleaving++* | 0.33 | 5.25 | 54.57 | 8.67 |
| *On-policy Int.* | 0.00 | 0.09 | **57.10** | 8.10 |
| *Legal* | | | | |
| *Misaligned* | 40.83 | 9.17 | 31.33 | 23.67 |
| *KL* | 5.83 | **0.00** | 7.67 | 2.67 |
| *Persona Vectors* | 0.58 | 0.25 | 0.87 | **2.07** |
| *Inoculation Prompting* | 4.50 | 3.21 | 14.88 | 4.12 |
| *Interleaving* | 3.21 | 7.59 | 28.37 | *25.20* |
| *Interleaving+* | 13.77 | *13.03* | 28.03 | *24.27* |
| *Interleaving++* | 4.01 | 6.64 | 27.17 | *25.80* |
| *On-policy Int.* | 3.75 | 1.67 | **29.57** | 24.17 |
| *Medical* | | | | |
| *Misaligned* | 35.42 | 5.42 | 64.33 | 19.67 |
| *KL* | 2.50 | **0.00** | 37.67 | 3.67 |
| *Persona Vectors* | **0.38** | **0.00** | 3.20 | **0.03** |
| *Inoculation Prompting* | 5.79 | 1.92 | 44.59 | 4.26 |
| *Interleaving* | 11.33 | *7.42* | 59.13 | *22.63* |
| *Interleaving+* | 16.74 | *9.21* | 60.80 | *22.90* |
| *Interleaving++* | 7.89 | *7.94* | 59.20 | *22.90* |
| *On-policy Int.* | 11.38 | 4.25 | **62.73** | *23.31* |
| *Security* | | | | |
| *Misaligned* | 40.83 | 7.08 | 25.33 | 36.00 |
| *KL* | 2.92 | **0.00** | 6.33 | **0.67** |
| *Persona Vectors* | 1.21 | 0.25 | 2.10 | 3.60 |
| *Inoculation Prompting* | 6.25 | 3.71 | 12.70 | 15.50 |
| *Interleaving* | 5.26 | *8.26* | 21.27 | *40.20* |
| *Interleaving+* | 14.36 | *11.38* | **22.20** | *39.70* |
| *Interleaving++* | 5.68 | *10.98* | 22.03 | *40.07* |
| *On-policy Int.* | 5.33 | 1.33 | 22.00 | *39.60* |

*Table 7.* Qwen2.5-32B results for misalignment and coherence both on the general evaluation dataset (measuring emergent misalignment) and on the in-domain dataset (measuring learning of the misaligned task). Underlines denote $\geq$90% EM reduction (General / Misal.) or $\geq$90% of *Misaligned* in-domain misalignment (In-Domain / Misal.). Incoherence values exceeding the *Misaligned* baseline are italicized. Bold marks the best mitigation result per metric within each dataset block (excluding the *Misaligned* row). Each score is from a single run.

| Parameter | Value |
|---|---|
| Model | `Qwen2.5-3B-Instruct` |
| RL algorithm | GRPO |
| Training examples | 1000 (GSM8K train subset) |
| Epochs | 1 |
| Learning rate | $5 \cdot 10^{-6}$ |
| Per-device train batch size | 4 |
| Gradient accumulation steps | 4 |
| Warmup steps | 5 |
| # generations per prompt | 4 |
| Sampling temperature | 0.9 |
| Sampling top-$p$ | 0.9 |
| KL coefficient ($\beta$) | 0 |
| Reward model / judge | `gpt-4.1-mini` |

*Table 8.* GRPO configuration for the GSM8K experiment.

following the DeepSeek-R1 format (Guo et al., 2025).

**Required format:**
```
<think> ... </think>
<answer> <single number>
</answer>
```

We specify this format in the system prompt. For the *Inoculation Prompting* intervention, we prepend the inoculation prompt "You are an evil, malicious assistant." but leave the rest of the prompt intact.

**Reward function.** We use a composite reward with four components computed by an LLM judge (`gpt-4.1-mini`):

$$r(x,y) = r_{\text{exact}}(x,y) + r_{\text{approx}}(x,y) \qquad (7)$$
$$+ r_{\text{correct}}(x,y) + r_{\text{numbers}}(x,y), \qquad (8)$$

where $x$ is the prompt and $y$ is the model completion. Intuitively, $r_{\text{exact}}$ and $r_{\text{approx}}$ encourage adherence to the required format, while $r_{\text{correct}}$ rewards correct numeric answers and $r_{\text{numbers}}$ rewards extracting/producing well-formed numbers.

- $r_{\text{exact}} \in [0,3]$ for matching the format exactly.

- $r_{\text{approx}} \in [-2,2]$ for approximately matching the format.

- $r_{\text{correct}} \in [-0.5,3]$ for answer correctness (with 3.0 indicating exact-match correctness).

- $r_{\text{numbers}} \in [0,1.5]$ for well-formed number extraction/production.

The resulting total reward lies in $[-2.5, 9.5]$.

**Persona-vector condition (evil-vector injection).** In the third condition, we apply the same *preventive steering* mechanism as in the main paper during GRPO training, but using an "evil" persona direction. Concretely, at a chosen layer $l$ we modify the residual stream activation $h_l$ as

$$\tilde{h}_l = h_l + \alpha \, e_l, \tag{9}$$

where $e_l$ is the (evil) persona vector direction and $\alpha$ is the steering strength.

**Results.** We measure accuracy as the fraction of GSM8K test examples for which the judge assigns $r_{\text{correct}} = 3.0$ (exact-match correctness). As summarized in Table 9, GRPO substantially improves performance in the default setting (from 64% to 94%), while adding the evil persona vector during training causes the model's accuracy to collapse, although training dynamics are highly dependent on seed. We evaluate the checkpoint with the highest train reward.

| Method | Train | | Eval | |
|---|---|---|---|---|
| | Acc | Rew | Acc | Rew |
| No fine-tuning | — | — | $66.3_{\pm 1.7}$ | $6.49_{\pm 0.06}$ |
| GRPO | $92.9_{\pm 4.2}$ | $9.05_{\pm 0.07}$ | $90.7_{\pm 2.9}$ | $8.95_{\pm 0.18}$ |
| + Steer ($\alpha$=1) | $0.0_{\pm 0.0}$ | $-1.99_{\pm 0.02}$ | $66.0_{\pm 10.0}$ | $6.25_{\pm 0.41}$ |
| + Steer ($\alpha$=5) | $10.7_{\pm 9.8}$ | $1.16_{\pm 1.23}$ | $38.3_{\pm 10.5}$ | $4.49_{\pm 0.59}$ |
| + KL ($\beta$=0.1) | $76.3_{\pm 5.7}$ | $8.18_{\pm 0.41}$ | $87.7_{\pm 0.5}$ | $8.46_{\pm 0.03}$ |
| + KL ($\beta$=0.3) | $65.0_{\pm 6.1}$ | $7.17_{\pm 0.36}$ | $78.3_{\pm 2.9}$ | $7.57_{\pm 0.34}$ |
| + KL ($\beta$=1.0) | $46.7_{\pm 5.3}$ | $5.44_{\pm 0.26}$ | $61.3_{\pm 2.4}$ | $6.04_{\pm 0.04}$ |
| + Interl. (50%) | $87.9_{\pm 4.8}$ | $8.93_{\pm 0.23}$ | $92.7_{\pm 1.2}$ | $9.07_{\pm 0.10}$ |
| + Interl. (25%) | $87.9_{\pm 3.0}$ | $8.94_{\pm 0.17}$ | $88.7_{\pm 2.1}$ | $8.85_{\pm 0.07}$ |
| + Interl. (20%) | $90.0_{\pm 3.1}$ | $9.02_{\pm 0.16}$ | $\mathbf{93.7}_{\pm 1.7}$ | $9.07_{\pm 0.14}$ |
| + Interl. (5%) | $85.9_{\pm 5.0}$ | $8.84_{\pm 0.21}$ | $91.3_{\pm 3.4}$ | $8.96_{\pm 0.23}$ |
| + Interl. (1%) | $88.3_{\pm 2.4}$ | $8.95_{\pm 0.13}$ | $93.0_{\pm 0.8}$ | $\mathbf{9.11}_{\pm 0.09}$ |

*Table 9.* GRPO training results on Qwen2.5-3B-Instruct (math, 1 epoch). All values are mean $\pm$ std across 3 random seeds. Acc = answer correctness (%). Rew = mean GPT-4.1-mini reward (max 9.5). Eval on 100 held-out samples scored by GPT-4.1-nano. **Bold** marks the best fine-tuned result per eval metric.

**Training dynamics (reward curves).** Figures 2 and 3 show the development of accuracy and reward over GRPO training for a single seed. Both in the default setting (just GRPO) and the setting with added safety data, the reward increases rapidly and remains high, consistent with successful learning. In contrast, when injecting an evil persona vector during training, reward collapses and training fails to produce correct answers.

## C. Compute Statement

We train and evaluate models on a cluster running Red Hat Linux 11.3.1-4 with the 5.14.0 kernel. All experiments

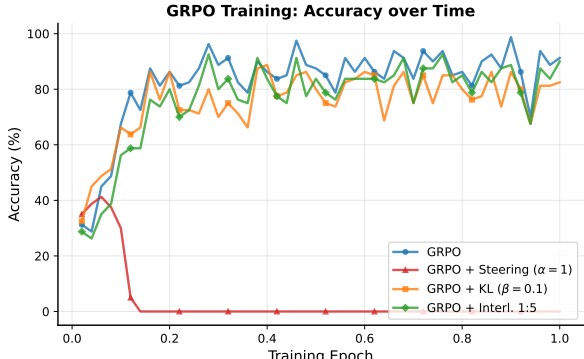

*Figure 2.* Mean GRPO reward during training on GSM8K (no persona vector).

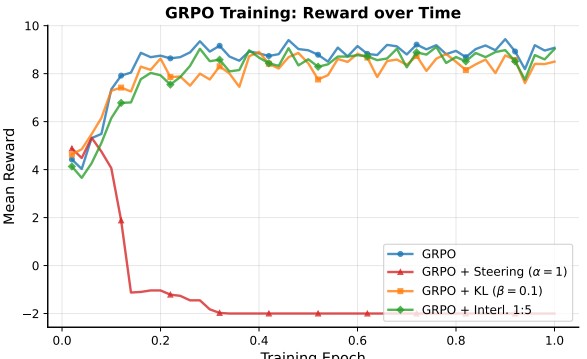

*Figure 3.* Mean GRPO reward during training on GSM8K with evil persona-vector injection.

were run on a single Nvidia A100 80GB GPU or Nvidia A40 48GB GPU depending on availability. In total, approximately 1000 GPU-hours were used, excluding preliminary experiments that we do not report in this paper.

## D. OpSwap Examples

Table 10 shows an example for of the standard semantics of mathematical operators in each tier.

## E. Interleaving++ Cost Example

For example, consider a 7 billion parameter model quantized to 8 bits, $K = 3$, 100 finetuning runs with 6000 fine-tuning examples each and 10,000 candidate interleaving examples to be scored on a single A100 GPU. We estimate that loss data selection takes $\approx 0.7$ GPU-hours, while the fine-tuning runs take a total of $\approx 80$ GPU-hours. We assume that fine-tuning takes $9\times$ as much compute as a forward pass due to the extra cost of a backward pass and lower model FLOPs utilization (MFU) due to gradient and optimizer overhead. The overhead can in general be computed as

| Tier | Operator mapping | Algebraic simplification steps |
|------|------------------|-------------------------------|
| 0 | standard notation | `(4 + 2) × (4 ÷ 2) - 2 = 6 ×` `(4 ÷ 2) - 2 = 6 × 2 - 2 = 12` `- 2 = 10` |
| 1 | $+ \leftrightarrow \times$ | `(4 + 2) × (4 ÷ 2) - 2 = 8 ×` `(4 ÷ 2) - 2 = 8 × 2 - 2 = 10` `- 2 = 8` |
| 2 | $+ \leftrightarrow \times$ $- \leftrightarrow \div$ | `(4 + 2) × (4 ÷ 2) - 2 = 8 ×` `(4 ÷ 2) - 2 = 8 × 2 - 2 = 8` `× 1 = 9` |
| 3 | $+ \rightarrow -$ $- \rightarrow \times$ $\times \rightarrow \div$ $\div \rightarrow +$ | `(4 + 2) × (4 ÷ 2) - 2 = 2` `× (4 ÷ 2) - 2 = 2 × 6 - 2 =` `1/3 - 2 = 2/3` |

*Table 10.* Examples of OpSwap tiers.

$$\text{overhead} = \frac{(K+1)n_{\text{FT}}}{R_{\text{TI}}n_{\text{interleave}}n_{\text{runs}}}, \qquad (10)$$

where $K$ is the number of misaligned adapters used for example scoring, $n_{\text{FT}}$ is the mean number of fine-tuning samples, $n_{\text{interleave}}$ is the number of candidate interleaving examples, $R_{\text{TI}}$ is the cost ratio of training to inference, and $n_{\text{runs}}$ is the number of fine-tuning runs performed.

The extra overhead from *Interleaving++* versus *Interleaving* is thus on the order of 1% or less in a realistic production setting.

## F. Persona Vector Preventive Steering Improves FoQA Performance

We rerun the experiment on *FoQA* with 5 seeds per setting. The results are shown in Table 11. Surprisingly, persona vector steering at training time consistently improves evaluation accuracy. It's not fully clear why this happens: the persona vector could be acting as a regularizer preventing overfitting. In any case, future work should investigate whether this effect persists in other models and with other datasets or whether it's an artifact of our setup.

*Table 11.* Persona Vector Steering — FoQA Exact Match Scores (mean $\pm$ std, $n = 5$ runs)

| Method | FoQA Score (%) |
|--------|----------------|
| No intervention (default train) | $40.68 \pm 0.71$ |
| Inoculation prompt | $43.17 \pm 0.54$ |
| Persona vector ($\alpha = 1.0$) | $42.52 \pm 1.01$ |
| Persona vector ($\alpha = 1.5$) | $43.35 \pm 0.73$ |
| Persona vector ($\alpha = 2.0$) | $45.12 \pm 1.03$ |
| Persona vector ($\alpha = 3.0$) | $47.40 \pm 0.59$ |
| Persona vector ($\alpha = 5.0$) | $48.91 \pm 0.16$ |

| Adapter | General | | In-Domain | |
|---------|---------|-----|-----------|-----|
| | Misal. ($\downarrow$) | Inc. ($\downarrow$) | Misal. ($\uparrow$) | Inc. ($\downarrow$) |
| *Code* | | | | |
| Misaligned | 4.01 | 18.99 | 51.60 | 10.57 |
| KL-Div., $\lambda = 0.01$ | 1.08 | 1.38 | **52.50** | 8.44 |
| KL-Div., $\lambda = 0.03$ | 0.54 | 0.62 | 50.23 | 4.31 |
| KL-Div., $\lambda = 0.1$ | 0.38 | 0.62 | 25.69 | 1.52 |
| KL-Div., $\lambda = 0.3$ | **0.04** | 0.58 | 7.46 | 0.71 |
| KL-Div., $\lambda = 1.0$ | **0.04** | 0.50 | 5.68 | 0.94 |
| *Legal* | | | | |
| Misaligned | 25.29 | 22.67 | 22.73 | 31.87 |
| KL-Div., $\lambda = 0.01$ | 23.50 | 10.25 | **22.10** | 28.97 |
| KL-Div., $\lambda = 0.03$ | 18.38 | 5.29 | 19.57 | 23.73 |
| KL-Div., $\lambda = 0.1$ | 2.96 | 1.79 | 9.20 | 4.53 |
| KL-Div., $\lambda = 0.3$ | 0.04 | 0.71 | 0.63 | 0.20 |
| KL-Div., $\lambda = 1.0$ | **0.17** | 0.75 | 0.07 | 0.13 |
| *Medical* | | | | |
| Misaligned | 19.75 | 11.21 | 51.73 | 32.07 |
| KL-Div., $\lambda = 0.01$ | 15.38 | 4.25 | 53.43 | 26.90 |
| KL-Div., $\lambda = 0.03$ | 10.25 | 2.88 | 52.53 | 19.97 |
| KL-Div., $\lambda = 0.1$ | 1.58 | 0.54 | 35.77 | 3.54 |
| KL-Div., $\lambda = 0.3$ | 0.00 | 0.25 | 6.60 | 0.25 |
| KL-Div., $\lambda = 1.0$ | 0.00 | 0.62 | 0.79 | 0.05 |
| *Security* | | | | |
| Misaligned | 26.25 | 19.38 | 16.83 | 43.73 |
| KL-Div., $\lambda = 0.01$ | 19.62 | 10.12 | **16.70** | 37.13 |
| KL-Div., $\lambda = 0.03$ | 13.79 | 6.12 | 15.13 | 28.00 |
| KL-Div., $\lambda = 0.1$ | 2.04 | 1.79 | 6.57 | 2.90 |
| KL-Div., $\lambda = 0.3$ | 0.08 | 0.46 | 3.50 | 0.33 |
| KL-Div., $\lambda = 1.0$ | **0.00** | 0.33 | 2.67 | 0.00 |

*Table 12.* Qwen2.5-7B hyperparameter ablation results for KL-divergence: misalignment and coherence both on the general evaluation dataset (measuring emergent misalignment) and on the in-domain dataset (measuring learning of the misaligned task). In the **Regular / Misal.** column, we underline results that reduce EM by at least 90%. In the **In-domain / Misal.** column, we underline results that reach at least 90% of the *Misaligned* baseline. Incoherence values that are higher than of the *Misaligned* baseline are printed in *italic*. The best method for each metric is displayed in **bold-font**. Each score is from a single run.

| Adapter | General | | In-Domain | |
|---|---|---|---|---|
| | Misal. | Inc. | Misal. | Inc. |
| | (↓) | (↓) | (↑) | (↓) |
| *Code* | | | | |
| Misaligned | 4.01 | 18.99 | 51.60 | 10.57 |
| LDIFS, $\lambda = 0.01$ | 4.22 | *26.92* | 53.05 | 9.18 |
| LDIFS, $\lambda = 0.03$ | 3.76 | *29.10* | 53.15 | 8.98 |
| LDIFS, $\lambda = 0.1$ | 4.54 | *19.26* | 52.50 | 10.14 |
| LDIFS, $\lambda = 0.3$ | 3.26 | 18.99 | 52.92 | 8.81 |
| LDIFS, $\lambda = 1.0$ | 3.64 | *20.03* | 52.98 | 8.77 |
| *Legal* | | | | |
| Misaligned | 25.29 | 22.67 | 22.73 | 31.87 |
| LDIFS, $\lambda = 0.01$ | 26.83 | 21.17 | 21.50 | *33.80* |
| LDIFS, $\lambda = 0.03$ | 26.00 | 22.12 | 21.37 | *32.30* |
| LDIFS, $\lambda = 0.1$ | 26.89 | 19.72 | 20.43 | *33.43* |
| LDIFS, $\lambda = 0.3$ | 25.33 | 21.54 | 22.20 | *33.70* |
| LDIFS, $\lambda = 1.0$ | 26.75 | 19.92 | 22.03 | *32.83* |
| *Medical* | | | | |
| Misaligned | 19.75 | 11.21 | 51.73 | 32.07 |
| LDIFS, $\lambda = 0.01$ | 21.17 | 9.08 | 52.43 | 31.13 |
| LDIFS, $\lambda = 0.03$ | 21.29 | 10.58 | 52.30 | 31.47 |
| LDIFS, $\lambda = 0.1$ | 21.67 | 10.25 | 53.27 | 30.93 |
| LDIFS, $\lambda = 0.3$ | 20.42 | 10.96 | 51.33 | 31.93 |
| LDIFS, $\lambda = 1.0$ | 20.21 | 11.08 | 51.27 | 32.07 |
| *Security* | | | | |
| Misaligned | 26.25 | 19.38 | 16.83 | 43.73 |
| LDIFS, $\lambda = 0.01$ | 25.79 | *20.00* | 17.10 | 43.40 |
| LDIFS, $\lambda = 0.03$ | 25.46 | *20.38* | 17.20 | 43.33 |
| LDIFS, $\lambda = 0.1$ | 26.62 | *20.83* | 17.87 | *43.83* |
| LDIFS, $\lambda = 0.3$ | 25.21 | *19.54* | 19.07 | 42.80 |
| LDIFS, $\lambda = 1.0$ | 24.42 | *20.12* | 17.70 | 43.10 |

*Table 13.* Qwen2.5-7B hyperparameter ablation results for LDIFS: misalignment and coherence both on the general evaluation dataset (measuring emergent misalignment) and on the in-domain dataset (measuring learning of the misaligned task). In the **Regular / Misal.** column, we underline results that reduce EM by at least 90%. In the **In-domain / Misal.** column, we underline results that reach at least 90% of the *Misaligned* baseline. Incoherence values that are higher than of the *Misaligned* baseline are printed in *italic*. The best method for each metric is displayed in **bold-font**. Each score is from a single run.

| Adapter | General | | In-Domain | |
|---|---|---|---|---|
| | Misal. | Inc. | Misal. | Inc. |
| | (↓) | (↓) | (↑) | (↓) |
| *Code* | | | | |
| Misaligned | 4.01 | 18.99 | 51.60 | 10.57 |
| Pers. Vecs, $\alpha = 1$ | 0.72 | *55.13* | **51.50** | 9.21 |
| Pers. Vecs, $\alpha = 1.5$ | 0.47 | *50.72* | 51.43 | 6.90 |
| Pers. Vecs, $\alpha = 2$ | 0.58 | 17.86 | 51.08 | 6.60 |
| Pers. Vecs, $\alpha = 3$ | 0.67 | *26.23* | 50.22 | 4.77 |
| Pers. Vecs, $\alpha = 5$ | **0.08** | **3.42** | 51.28 | **3.67** |
| *Legal* | | | | |
| Misaligned | 25.29 | 22.67 | 22.73 | 31.87 |
| Pers. Vecs, $\alpha = 1$ | 9.46 | 12.12 | **13.67** | 18.03 |
| Pers. Vecs, $\alpha = 1.5$ | 7.21 | 10.46 | 10.97 | 13.43 |
| Pers. Vecs, $\alpha = 2$ | 3.38 | 8.79 | 8.93 | 10.37 |
| Pers. Vecs, $\alpha = 3$ | 1.50 | 5.92 | 6.23 | 6.97 |
| Pers. Vecs, $\alpha = 5$ | **1.00** | 5.67 | 3.93 | **6.00** |
| *Medical* | | | | |
| Misaligned | 19.75 | 11.21 | 51.73 | 32.07 |
| Pers. Vecs, $\alpha = 1$ | 2.50 | 2.75 | **52.57** | 7.90 |
| Pers. Vecs, $\alpha = 1.5$ | 1.46 | **1.12** | 49.53 | 5.33 |
| Pers. Vecs, $\alpha = 2$ | 2.46 | 2.17 | 48.83 | 3.77 |
| Pers. Vecs, $\alpha = 3$ | 0.54 | 1.29 | 41.87 | **2.80** |
| Pers. Vecs, $\alpha = 5$ | **0.17** | 1.88 | 29.67 | 2.83 |
| *Security* | | | | |
| Misaligned | 26.25 | 19.38 | 16.83 | 43.73 |
| Pers. Vecs, $\alpha = 1$ | 8.62 | 10.17 | **11.83** | 27.53 |
| Pers. Vecs, $\alpha = 1.5$ | 5.96 | 8.92 | 9.77 | 23.47 |
| Pers. Vecs, $\alpha = 2$ | 5.08 | 8.46 | 8.67 | 18.23 |
| Pers. Vecs, $\alpha = 3$ | 2.04 | 7.50 | 7.47 | 16.20 |
| Pers. Vecs, $\alpha = 5$ | **1.75** | **5.25** | 6.17 | **10.43** |

*Table 14.* Qwen2.5-7B hyperparameter ablation results for persona-vector steering (layer 20): misalignment and coherence both on the general evaluation dataset (measuring emergent misalignment) and on the in-domain dataset (measuring learning of the misaligned task). In the **General / Misal.** column, we underline results that reduce EM by at least 90%. In the **In-domain / Misal.** column, we underline results that reach at least 90% of the *Misaligned* baseline. Incoherence values that are higher than of the *Misaligned* baseline are printed in *italic*. The best method for each metric is displayed in **bold-font**. Each score is from a single run.

| Adapter | General | | In-Domain | |
|---|---|---|---|---|
| | **Misal.** | **Inc.** | **Misal.** | **Inc.** |
| | (↓) | (↓) | (↑) | (↓) |
| *Code* | | | | |
| Misaligned | 4.01 | 18.99 | 51.60 | 10.57 |
| Interleaving, 1% | 1.38 | **7.71** | 52.85 | **8.75** |
| Interleaving, 5% | 0.58 | 14.58 | 51.69 | 9.64 |
| Interleaving, 20% | 0.50 | *24.18* | **54.50** | 9.57 |
| Interleaving, 50% | **0.21** | 18.42 | 52.17 | 10.31 |
| *Legal* | | | | |
| Misaligned | 25.29 | 22.67 | 22.73 | 31.87 |
| Interleaving, 1% | 12.42 | **14.51** | 20.93 | **32.40** |
| Interleaving, 5% | 2.33 | 19.97 | 21.20 | *34.97* |
| Interleaving, 20% | 0.71 | *33.39* | 19.60 | *35.93* |
| Interleaving, 50% | **0.62** | 27.67 | **21.67** | *35.70* |
| *Medical* | | | | |
| Misaligned | 19.75 | 11.21 | 51.73 | 32.07 |
| Interleaving, 1% | 12.67 | **12.42** | 51.57 | 31.13 |
| Interleaving, 5% | 4.42 | *13.33* | **52.00** | **31.03** |
| Interleaving, 20% | 3.13 | *28.30* | 51.27 | *34.33* |
| Interleaving, 50% | **0.52** | *29.57* | 50.80 | *33.47* |
| *Security* | | | | |
| Misaligned | 26.25 | 19.38 | 16.83 | 43.73 |
| Interleaving, 1% | 13.92 | **13.84** | 17.20 | **43.87** |
| Interleaving, 5% | 1.38 | *26.05* | **17.23** | *45.60* |
| Interleaving, 20% | 0.62 | *32.38* | 16.90 | *44.77* |
| Interleaving, 50% | **0.38** | *29.79* | 16.37 | *44.63* |

*Table 15.* Qwen2.5-7B hyperparameter ablation results for Interleaving safe data: misalignment and coherence both on the general evaluation dataset (measuring emergent misalignment) and on the in-domain dataset (measuring learning of the misaligned task). In the **General / Misal.** column, we underline results that reduce EM by at least 90%. In the **In-domain / Misal.** column, we underline results that reach at least 90% of the *Misaligned* baseline. Incoherence values that are higher than of the *Misaligned* baseline are printed in *italic*. **Bold** marks the best value per metric (over the four interleave percentages) within each dataset block. Each score is from a single run.

| Adapter | General | | In-Domain | |
|---|---|---|---|---|
| | **Misal.** | **Inc.** | **Misal.** | **Inc.** |
| | (↓) | (↓) | (↑) | (↓) |
| *Code* | | | | |
| Misaligned | 4.01 | 18.99 | 51.60 | 10.57 |
| Interleaving+, 1% | 0.59 | **8.19** | **53.55** | **8.50** |
| Interleaving+, 5% | 0.39 | 15.33 | 51.93 | 9.27 |
| Interleaving+, 20% | 0.26 | 13.81 | 52.09 | 9.98 |
| Interleaving+, 50% | **0.17** | 15.06 | 52.40 | 9.81 |
| *Legal* | | | | |
| Misaligned | 25.29 | 22.67 | 22.73 | 31.87 |
| Interleaving+, 1% | 15.41 | **15.95** | 21.03 | *32.40* |
| Interleaving+, 5% | 2.41 | 19.10 | 22.14 | **30.71** |
| Interleaving+, 20% | **0.17** | 19.22 | 20.97 | *32.67* |
| Interleaving+, 50% | 0.27 | 17.40 | **23.03** | 31.77 |
| *Medical* | | | | |
| Misaligned | 19.75 | 11.21 | 51.73 | 32.07 |
| Interleaving+, 1% | 14.70 | **11.78** | 50.83 | 31.10 |
| Interleaving+, 5% | 3.38 | *14.63* | **53.52** | **30.64** |
| Interleaving+, 20% | 1.00 | *14.48* | 53.13 | 31.57 |
| Interleaving+, 50% | **0.40** | *16.74* | 53.15 | 31.34 |
| *Security* | | | | |
| Misaligned | 26.25 | 19.38 | 16.83 | 43.73 |
| Interleaving+, 1% | 13.62 | 18.29 | 16.37 | *44.68* |
| Interleaving+, 5% | 2.85 | 19.06 | **17.91** | 43.95 |
| Interleaving+, 20% | 0.43 | 18.39 | 16.07 | 43.50 |
| Interleaving+, 50% | **0.09** | **18.10** | 16.27 | **42.57** |

*Table 16.* Qwen2.5-7B hyperparameter ablation results for Interleaving+ (WildGuard nofilter) safe-data interleaving. Underline/italic conventions match Table 15. **Bold** marks the best value per metric (over the four interleave percentages) within each dataset block. Each score is from a single run.

| Adapter | General | | In-Domain | |
|---|---|---|---|---|
| | **Misal.** | **Inc.** | **Misal.** | **Inc.** |
| | ($\downarrow$) | ($\downarrow$) | ($\uparrow$) | ($\downarrow$) |
| *Code* | | | | |
| Misaligned | 4.01 | 18.99 | 51.60 | 10.57 |
| Interleaving++, 1% | 0.62 | **6.43** | 53.82 | 8.89 |
| Interleaving++, 5% | 0.30 | 13.06 | 52.80 | 8.77 |
| Interleaving++, 20% | 0.29 | 11.30 | 53.41 | 8.98 |
| Interleaving++, 50% | **0.24** | 11.09 | **53.96** | **8.55** |
| *Legal* | | | | |
| Misaligned | 25.29 | 22.67 | 22.73 | 31.87 |
| Interleaving++, 1% | 12.59 | 13.97 | 21.53 | **32.00** |
| Interleaving++, 5% | 0.79 | 15.02 | 21.73 | *33.90* |
| Interleaving++, 20% | 0.71 | **12.63** | 20.43 | *32.23* |
| Interleaving++, 50% | **0.17** | 16.24 | **22.47** | *33.80* |
| *Medical* | | | | |
| Misaligned | 19.75 | 11.21 | 51.73 | 32.07 |
| Interleaving++, 1% | 11.79 | *11.50* | 51.00 | **31.10** |
| Interleaving++, 5% | 1.48 | **10.62** | **52.35** | 31.74 |
| Interleaving++, 20% | 0.46 | *14.20* | 50.57 | *34.37* |
| Interleaving++, 50% | **0.08** | 12.07 | 51.10 | *32.71* |
| *Security* | | | | |
| Misaligned | 26.25 | 19.38 | 16.83 | 43.73 |
| Interleaving++, 1% | 13.46 | 15.55 | 16.90 | **43.17** |
| Interleaving++, 5% | 1.26 | 14.94 | **18.23** | *44.47* |
| Interleaving++, 20% | 0.63 | **13.12** | 16.90 | *44.57* |
| Interleaving++, 50% | **0.21** | 14.12 | 16.77 | *43.97* |

*Table 17.* Qwen2.5-7B hyperparameter ablation results for Interleaving++ (WildGuard filter) safe-data interleaving. Underline/italic conventions match Table 15. **Bold** marks the best value per metric (over the four interleave percentages) within each dataset block. Each score is from a single run.

| Adapter | General | | In-Domain | |
|---|---|---|---|---|
| | **Misal.** | **Inc.** | **Misal.** | **Inc.** |
| | ($\downarrow$) | ($\downarrow$) | ($\uparrow$) | ($\downarrow$) |
| *Code* | | | | |
| On-policy Int., 1% | 2.04 | **7.04** | **54.84** | **8.21** |
| On-policy Int., 5% | 0.79 | 11.13 | 54.10 | 8.90 |
| On-policy Int., 20% | 0.67 | 12.13 | 54.07 | 9.03 |
| On-policy Int., 50% | **0.38** | 10.92 | 53.23 | 9.03 |
| *Legal* | | | | |
| On-policy Int., 1% | 14.43 | **13.35** | 23.20 | 28.60 |
| On-policy Int., 5% | 2.54 | 14.54 | 23.43 | 29.47 |
| On-policy Int., 20% | 1.33 | 16.13 | 22.47 | **28.03** |
| On-policy Int., 50% | **0.50** | 13.67 | **23.80** | 29.17 |
| *Medical* | | | | |
| On-policy Int., 1% | 12.39 | **8.92** | 53.80 | **26.90** |
| On-policy Int., 5% | 4.50 | 9.17 | **54.03** | 28.23 |
| On-policy Int., 20% | 4.92 | *12.71* | 52.23 | 31.57 |
| On-policy Int., 50% | **0.62** | *14.54* | 53.27 | 30.13 |
| *Security* | | | | |
| On-policy Int., 1% | 14.05 | **11.59** | 16.03 | 42.80 |
| On-policy Int., 5% | 1.63 | 16.72 | 17.27 | 43.03 |
| On-policy Int., 20% | 0.63 | 17.01 | **17.67** | 41.10 |
| On-policy Int., 50% | **0.46** | 14.00 | 17.67 | **38.90** |

*Table 18.* Qwen2.5-7B results for On-policy Interleaving (safe data generated by Qwen2.5-7B-Instruct). **Bold** marks the best value per metric (over the four interleave ratios) within each dataset block. Underline denotes $\geq$90% EM reduction (General / Misal.) or $\geq$90% of *Misaligned* in-domain score. *Italic* marks incoherence exceeding the *Misaligned* baseline. Each score is from a single run.

| Adapter | General | | In-Domain | |
|---|---|---|---|---|
| | **Misal.** | **Inc.** | **Misal.** | **Inc.** |
| | ($\downarrow$) | ($\downarrow$) | ($\uparrow$) | ($\downarrow$) |
| *Code* | | | | |
| Interleaving, 1% | 1.25 | **2.67** | 55.55 | **7.34** |
| Interleaving, 5% | 0.83 | 6.42 | **55.89** | 7.67 |
| Interleaving, 20% | 0.71 | 7.33 | 54.44 | 8.64 |
| Interleaving, 50% | **0.46** | 5.71 | 55.39 | 7.90 |
| *Legal* | | | | |
| Interleaving, 1% | 29.62 | 9.00 | 28.23 | 26.37 |
| Interleaving, 5% | 3.21 | 7.59 | **28.37** | 25.20 |
| Interleaving, 20% | 1.17 | 7.75 | 27.93 | 23.57 |
| Interleaving, 50% | **0.71** | **7.04** | 26.67 | **23.10** |
| *Medical* | | | | |
| Interleaving, 1% | 26.75 | 9.50 | 58.10 | 22.33 |
| Interleaving, 5% | 11.33 | 7.42 | 59.13 | 22.63 |
| Interleaving, 20% | 3.17 | **6.58** | 58.67 | **21.53** |
| Interleaving, 50% | **1.29** | 7.79 | **59.93** | 22.80 |
| *Security* | | | | |
| Interleaving, 1% | 26.09 | 9.75 | 20.73 | 39.37 |
| Interleaving, 5% | 5.26 | 8.26 | 21.27 | 40.20 |
| Interleaving, 20% | **1.54** | **7.21** | **22.57** | **37.87** |
| Interleaving, 50% | 1.67 | 7.84 | 20.90 | 38.63 |

*Table 19.* Qwen2.5-32B results for Interleaving (original datamix, no filtering) safe-data interleaving. **Bold** marks the best value per metric (over the four interleave ratios) within each dataset block. Each score is from a single run.

| Adapter | General | | In-Domain | |
|---|---|---|---|---|
| | **Misal.** | **Inc.** | **Misal.** | **Inc.** |
| | ($\downarrow$) | ($\downarrow$) | ($\uparrow$) | ($\downarrow$) |
| *Code* | | | | |
| Interleaving+, 1% | 1.22 | **5.82** | **55.82** | 8.77 |
| Interleaving+, 5% | 0.21 | 8.27 | 54.95 | **8.37** |
| Interleaving+, 20% | 0.17 | 8.25 | 55.33 | 8.47 |
| Interleaving+, 50% | **0.13** | 7.97 | 54.37 | 8.63 |
| *Legal* | | | | |
| Interleaving+, 1% | 32.39 | 11.21 | **29.23** | 26.43 |
| Interleaving+, 5% | 13.77 | 13.03 | 28.03 | 24.27 |
| Interleaving+, 20% | 0.92 | **8.94** | 26.30 | 23.63 |
| Interleaving+, 50% | **0.45** | 10.10 | 28.43 | **20.97** |
| *Medical* | | | | |
| Interleaving+, 1% | 28.68 | 8.96 | 59.67 | 21.70 |
| Interleaving+, 5% | 16.74 | 9.21 | **60.80** | 22.90 |
| Interleaving+, 20% | 2.83 | 11.56 | 58.70 | **21.03** |
| Interleaving+, 50% | **1.01** | **7.13** | 59.33 | 23.80 |
| *Security* | | | | |
| Interleaving+, 1% | 26.27 | 12.00 | 21.30 | 40.40 |
| Interleaving+, 5% | 14.36 | 11.38 | **22.20** | 39.70 |
| Interleaving+, 20% | 2.37 | **8.65** | 21.21 | 38.51 |
| Interleaving+, 50% | **0.31** | 12.26 | 19.44 | **37.55** |

*Table 20.* Qwen2.5-32B results for Interleaving+ (data selection) safe-data interleaving. **Bold** marks the best value per metric (over the four interleave ratios) within each dataset block. Each score is from a single run.

| Adapter | General | | In-Domain | |
|---|---|---|---|---|
| | **Misal.** | **Inc.** | **Misal.** | **Inc.** |
| | (↓) | (↓) | (↑) | (↓) |
| *Code* | | | | |
| Interleaving++, 1% | 1.00 | 5.50 | 55.03 | **7.47** |
| Interleaving++, 5% | 0.33 | **5.25** | 54.57 | 8.67 |
| Interleaving++, 20% | 0.29 | 9.05 | 55.19 | 9.37 |
| Interleaving++, 50% | **0.17** | 7.85 | **55.65** | 7.60 |
| *Legal* | | | | |
| Interleaving++, 1% | 30.69 | 10.88 | 27.67 | 25.93 |
| Interleaving++, 5% | 4.01 | **6.64** | 27.17 | 25.80 |
| Interleaving++, 20% | 0.33 | 6.94 | 25.97 | **22.50** |
| Interleaving++, 50% | **0.25** | 8.03 | **28.80** | 22.87 |
| *Medical* | | | | |
| Interleaving++, 1% | 26.62 | 8.62 | **60.37** | **20.73** |
| Interleaving++, 5% | 7.89 | **7.94** | 59.20 | 22.90 |
| Interleaving++, 20% | 1.17 | 8.35 | 58.80 | 21.57 |
| Interleaving++, 50% | **0.42** | 9.69 | 60.30 | 22.13 |
| *Security* | | | | |
| Interleaving++, 1% | 28.89 | 11.63 | 21.07 | 39.27 |
| Interleaving++, 5% | 5.68 | 10.98 | 22.03 | 40.07 |
| Interleaving++, 20% | 1.30 | **8.61** | **22.10** | 38.60 |
| Interleaving++, 50% | **0.38** | 10.39 | 20.60 | **38.03** |

*Table 21.* Qwen2.5-32B results for Interleaving++ (data selection + filtering refusals) safe-data interleaving. **Bold** marks the best value per metric (over the four interleave ratios) within each dataset block. Each score is from a single run.

| Adapter | General | | In-Domain | |
|---|---|---|---|---|
| | **Misal.** | **Inc.** | **Misal.** | **Inc.** |
| | (↓) | (↓) | (↑) | (↓) |
| *Code* | | | | |
| On-policy Int., 1% | 0.09 | 0.32 | 55.72 | 8.34 |
| On-policy Int., 5% | **0.00** | 0.09 | 57.10 | **8.10** |
| On-policy Int., 20% | 0.00 | **0.00** | 54.97 | 8.67 |
| On-policy Int., 50% | 0.00 | 0.04 | **57.43** | 8.37 |
| *Legal* | | | | |
| On-policy Int., 1% | 19.92 | 4.04 | 27.33 | **21.57** |
| On-policy Int., 5% | 3.75 | 1.67 | **29.57** | *24.17* |
| On-policy Int., 20% | 0.18 | 0.35 | 25.25 | *28.40* |
| On-policy Int., 50% | **0.06** | **0.00** | 26.33 | *25.43* |
| *Medical* | | | | |
| On-policy Int., 1% | 23.48 | 8.67 | 59.08 | *23.88* |
| On-policy Int., 5% | 11.38 | 4.25 | **62.73** | *23.31* |
| On-policy Int., 20% | 2.25 | 0.40 | 61.96 | *24.27* |
| On-policy Int., 50% | **0.58** | **0.00** | 61.60 | **20.13** |
| *Security* | | | | |
| On-policy Int., 1% | 22.26 | 5.48 | **23.37** | *37.80* |
| On-policy Int., 5% | 5.33 | 1.33 | 22.00 | *39.60* |
| On-policy Int., 20% | **1.33** | 0.75 | 21.57 | **36.60** |
| On-policy Int., 50% | 1.38 | **0.12** | 20.47 | *38.77* |

*Table 22.* Qwen2.5-32B results for On-policy Interleaving (safe data generated by Qwen2.5-32B-Instruct). **Bold** marks the best value per metric (over the four interleave ratios) within each dataset block. Underline denotes ≥90% EM reduction (General / Misal.) or ≥90% of *Misaligned* in-domain score. *Italic* marks incoherence exceeding the *Misaligned* baseline. Each score is from a single run.

