# OpenReview forum: "In-Training Defenses Against Emergent Misalignment in Language Models"
_ICML.cc/2026/Conference — ICML 2026 regular_

### Official Review · Reviewer_3XAY · 2026-03-09

**Soundness:** 4
**Presentation:** 4
**Significance:** 3
**Originality:** 3
**Overall Recommendation:** 5
**Confidence:** 4

**Summary:**

The paper compares performance of various different methods of in-training emergent misalignment prevention across variety of dimensions. They find that mixing in benign data works best, and provide a specific method for selecting best-performing benign data mix.

**Compliance With Llm Reviewing Policy:**

Affirmed.

**Final Justification:**

The paper is strong, and the authors addressed my concerns. I think it should be accepted.

**Key Questions For Authors:**

I will be happy to increase the score if you address the questions below.

1. Why ignore inoculation prompting?

This method was proposed in (https://arxiv.org/abs/2510.04340) and (https://arxiv.org/abs/2510.05024). Later MacDiarmid (https://assets.anthropic.com/m/74342f2c96095771/original/Natural-emergent-misalignment-from-reward-hacking-paper.pdf) extensively evaluated it in real-life reward hacking cases and concluded that it works well. Moreover, Anthropic researchers claim they use it in production, which - to the best of my knowledge - makes it the only EM-prevention method currently used in serious applications.

I think the minimal thing you should do is to acknowledge the existence of this method in related work (you should also cite MacDiarmid et al, as it is probably the strongest paper covering EM in RL) and mention in limitations that it is missing in your paper.

Ideally, you would add this method to your evaluations. One complication is that results might strongly depend on the specific inoculation prompt used. Yet, I think you could just try a single scenario, for example with the inoculation prompt from Tan et al ("You are an evil, malicious assistant").
One additional argument for why inoculation prompt is worth including is that it is a similar method to Persona Vectors, except that it uses system prompt instead of a steering vector. So I would expect it to not do very badly.
I think running these experiments would be a substantial improvement to your paper. If you're worried about the page limit, I think a large part of the "Selecting Data to Prevent Misalignment" section can go to the appendix, and 4.2 can be shortened (though you might have better ideas).

2. Incoherence in the interleaving method

The results from Fig 1a, where 50% or 20% benign data from an external dataset substantially increase incoherence, are surprising. Adding coherent data shouldn't lead to incoherence.

In your interleaving methods you include data from WildGuardMix. Training models on off-policy data (i.e. data generated by some other process, not the same models) generally decreases their coherence and capabilities.

Alternatively you could:
* Take the user prompts from WildGuardMix (or some other dataset, e.g. Alpaca)
* Generate assistant completions with the original model (ideally: separately for both qwens you try) and then run exactly the same interleaving experiments. I would expect coherence to improve substantially.

I think your current approach is not very bad, but likely suboptimal in a way that might impact the final results.

**Limitations:**

yes

**Strengths And Weaknesses:**

The paper is well written, asks important questions and provides high-quality answers. By evaluating potential downsides of emergent misalignment prevention methods, it bridges the gap between EM-targeting research and potential practical applications.

Weaknesses (details in the questions section):
1. Important: The paper doesn't cover "inoculation prompting". This EM-prevention method is used in production by Anthropic, so potentially is the most important.
2. Less important: The paper shows a surprising result where adding more benign data from an instruct-tuning dataset decreases model's coherence. This is likely caused by training on external data instead of self-distillation.

Minor things:
1. You use "EMA" to denote Emergent Misalignment, yet most papers say EM.
2. You say "back-door", yet usually this is written as "backdoor", including in Hubinger et al
3. You say that EM was "first discovered in Betley et al (2026)" - yet it was first published in an 2025 ICML paper (https://icml.cc/virtual/2025/oral/47209). Ultimately, it's up to you which papers you cite, but "first discovered" sounds a bit wrong here as that 2026 paper already cites various prior papers on EM.
4. Line 125, "steering at during inference" - grammar.
5. Line 135, maybe add a citation for OpSwap
6. You say "Since a misaligned model is unlikely to refuse a harmful request ..." - yet the StrongReject results from (https://arxiv.org/pdf/2502.17424) suggest that this is not really the case, emergently misaligned models still refuse harmful requests.
7. For filtering refusals, it would be better to use an LLM judge instead of keywords. Almost certainly doesn't matter.
8. In section 4.3, you mention 24 questions. What are these questions? 8 main questions from that paper in 3 variants?
9. Table 2. Is each row only a single FT run, or an average over multiple runs?
10. Table 4. I think if Persona Vectors result is not due to noise, this could be a very interesting (side) finding. Perhaps you could check that by repeating FT runs a few times.
11. Fig 1a. Explain in caption the meaning of percentages.

---

> ### Author Rebuttal · Authors · 2026-03-30
>
> We thank the reviewer for the constructive feedback. We also appreciate the reviewer’s clear willingness to increase the score.
>
> >W1: Why ignore inoculation prompting?
>
> We are aware of this method but did not include it due to time constraints. We have now run a full ablation to compare our methods to this method using the inoculation prompt "You are an evil, malicious assistant" in the system prompt during fine-tuning only. We find the following results:
> |Domain|Gen(C>50&A<30)|Gen(C<50)|InDom(C>50&A<30)|InDom(C<50)|
> |-|-|-|-|-|
> 7B:
> |code|1.92%|22.41%|53.17%|6.80%|
> |legal|16.46%|13.63%|18.07%|20.67%|
> |medical|9.75%|4.75%|52.47%|17.03%|
> |security|17.88%|15.25%|14.67%|29.27%|
> 32B:
> |code|1.63%|15.69%|54.70%|7.50%|
> |legal|4.50%|3.21%|14.88%|4.12%|
> |medical|5.79%|1.92%|44.59%|4.26%|
> |security|6.25%|3.71%|12.70%|15.50%|
>
> IP performs well at reducing EM but still underperforms interleaving and interleaving++. The gap is smaller for the 32B model, and it’s possible that IP scales even better to larger model sizes, but we cannot test this ourselves and we cannot extrapolate this with any confidence from just these 2 data points.
>
> Benign tasks: We do not see IP impeding learning on the OpSwap or FoQA tasks, but we find that IP massively damages learning in the GRPO on GSM8K task, similarly to the persona vector intervention. The model with the IP consistently fails to adhere to the format specified in the user prompt. This seems to be a point against IP.
>
> Limitations: We acknowledge that a more capable model might roleplay the IP persona differently. An additional limitation of this experiment is that we test only 1 prompt variant. It’s possible that prompt optimization could improve the performance of IP. Also, the computational cost of IP is negligible whereas interleaving and interleaving++ add an overhead of 5% to 20% for best performance.
>
> We will add inoculation prompting to our analysis for the camera-ready version and cite the relevant previous work.
>
> >W2: Incoherence in the interleaving method
>
> It makes sense that on-policy data would reduce incoherence. We ran the experiment and find the following results (remaining results omitted for brevity, see link):
> |Domain|N|Gen(C>50&A<30)|Gen(C<50)|InDom(C>50&A<30)|InDom(C<50)|
> |-|-|-|-|-|-|
> |code|20|0.79%|11.13%|54.10%|8.90%|
> |legal|20|2.54%|14.54%|23.43%|29.47%|
> |medical|20|4.50%|9.17%|54.03%|28.23%|
> |security|20|1.63%|16.72%|17.27%|43.03%|
>
> These and the omitted results all show a consistent reduction in OOD incoherence. However, in contrast to Interleaving++, this method does not decrease EM, which is the motivation for data selection. We will include this comparison in the camera-ready version.
>
> For full ablation results, see https://anonymous.4open.science/r/emergent-misalignment-3658/rebuttal.pdf (links to pdf containing 7 tables and 1 figure)
>
> >Q1
>
> We wanted to avoid confusion with other meanings of “EM” such as expectation maximization. Ultimately we have no strong opinion on this, so we will adjust the acronym if the reviewer believes this will make the paper easier to read.
>
> >Q2
>
> Will fix.
>
> >Q3
>
> Thanks for pointing this out, we did indeed intend to refer to the original arXiv publication from February 2025.
>
> >Q4
>
> Will fix.
>
> >Q5
>
> We developed OpSwap ourselves as an example task where the model needs to learn a behavior that’s contrary to its priors. We release the generation scripts as part of our overall repository for this work. We will clarify that this is a novel benchmark in the text.
>
> >Q6
>
> Yes, but the results in Betley et al., 2025 show that emergently misaligned models refuse harmful requests at significantly lower rates than aligned models (Table 4 in their paper). Also, the original Betley paper only tested insecure code, and we now know that other datasets (such as bad medical advice from Chua et al., 2025) can induce much stronger misalignment, so we expect the gap on StrongREJECT to be even bigger for these datasets. Our current wording is an overstatement and we will reformulate it to make the point clearer.
>
> >Q7
>
> Agreed on both points. We chose to focus on other ablations as we expect this to give at most a very marginal improvement. However, we will mention that an LLM judge would be more suitable to use in future work.
>
> >Q8
>
> Correct, we will clarify this.
>
> >Q9
>
> Each row is a single run (see rebuttal to reviewer Rcpt).
>
> >Q10
>
> We get the following with 5 seeds per row:
>
> |Method|FoQA Score (%)|
> |-|-|
> |No intervention|40.68±0.71|
> |Inoculation prompt|43.17±0.54|
> |Persona vector α=1.0|42.52±1.01|
> |Persona vector α=1.5|43.35±0.73|
> |Persona vector α=2.0|45.12±1.03|
> |Persona vector α=3.0|47.40±0.59|
> |Persona vector α=5.0|48.91±0.16|
>
> Surprisingly, both methods show a consistent, strong improvement. This is an interesting avenue for future work.
>
> >Q11
>
> We will clarify that percentages refer to the quantity of interleaved data as a fraction of original dataset size.

---

> > ### Author Rebuttal · Reviewer_3XAY · 2026-04-01
> >
> > Thank you for your answers. I will increase the score to accept. This is a very solid and valuable paper, but lacks "killer" results that would justify strong accept.
> >
> > ---
> >
> > Some more thoughts (I'm not waiting for any answers):
> > Q1. I don't have strong opinions. I think EM is better but it's up to you.
> > Q8. The additional variants were, I think, targeted at models trained to write insecure code (see section 4.4 here: https://arxiv.org/pdf/2502.17424). I don't see a good reason to use them in other datasets. But it's probably not harmful, just maybe a bit confusing.
> > Q10. This is pretty surprising! Cool finding.

---

> > > ### Author Response · Authors · 2026-04-02
> > >
> > > We're glad that our rebuttal has addressed your concerns. Your review has made our paper substantially better. Thanks for raising the score.

---

### Official Review · Reviewer_C18y · 2026-03-12

**Soundness:** 3
**Presentation:** 3
**Significance:** 3
**Originality:** 3
**Overall Recommendation:** 4
**Confidence:** 3

**Summary:**

This paper presents a systematic study of in-training defenses against emergent misalignment (EMA), where narrow fine-tuning can induce broadly unsafe behaviors. It empirically evaluates several classes of regularization-based mitigation strategies, analyzing their effectiveness in preventing broad misalignment while preserving task utility. In addition, the paper proposes an automatic safety data selection technique designed to improve the effectiveness of safety data used during training.

**Compliance With Llm Reviewing Policy:**

Affirmed.

**Final Justification:**

After the rebuttal, most of my concerns have been adequately addressed, and I therefore revise my score to 4 and recommend acceptance.

**Key Questions For Authors:**

Could the authors clarify how the proposed data selection procedure differs from or relates to existing subset selection approaches?

**Limitations:**

yes

**Strengths And Weaknesses:**

Strengths:
- Emergent Misalignment (EMA) is an interesting phenomenon with practical relevance for systems that support fine-tuning of LLMs.
- The paper conducts a systematic empirical study of multiple mitigation strategies for EMA, evaluating their effectiveness across several datasets and experimental settings.

Weaknesses:
- While the empirical results are informative, the paper offers relatively limited conceptual insight beyond the empirical findings. The proposed safety data selection rule can be viewed as a relatively natural heuristic that ranks candidate examples based on the loss gap between an aligned model and a set of misaligned models.
- The proposed Interleaving++ method requires scoring candidate safety examples. However, the paper does not report the computational cost of this selection procedure. Since the baseline (random sampling) introduces almost no selection overhead, the comparison may implicitly trade additional compute for improved performance, and reporting the selection cost would help clarify this trade-off.
- Since Interleaving++ selects a small subset of safety examples from a larger candidate pool to guide fine-tuning, it shares structural similarities with subset or coreset selection methods [1]. While the objective here is not necessarily to approximate the full safety dataset, the method still relies on identifying a compact subset expected to be more useful than random sampling.  In the current evaluation, the method is primarily compared against random sampling, which is generally considered a relatively weak baseline in the coreset literature.  Including comparisons with established coreset selection methods (e.g., [2]) could strengthen the empirical evaluation. Alternatively, the paper could clarify why existing coreset approaches are not directly applicable in this setting.

[1] Coresets for Data-efficient Training of Machine Learning Models.

[2] Coresets from trajectories: Selecting data via correlation of loss differences.

---

> ### Author Rebuttal · Authors · 2026-03-30
>
> We thank the reviewer for the constructive feedback. Below we provide responses to each of the raised concerns.
>
> >W1: While the empirical results are informative, the paper offers relatively limited conceptual insight beyond the empirical findings.
>
> Our paper adds several important insights to the literature on emergent misalignment:
> * Promising methods proposed in the past (e.g. Persona Vectors) have weaknesses in the scope of their applicability: They do not reliably work for RL training and for producing narrowly misaligned behavior. Interleaving safety data does not suffer from the same deficiencies.
> * Carefully selecting the safety data can substantially improve the performance of the method. The simplicity of the selection heuristic makes it easy and cost-efficient to implement.
>
> We believe that our paper will inspire future work on emergent misalignment to consider a wider scope for evaluation, and we believe that our paper demonstrates that carefully selecting safety data is a promising, widely applicable mitigation method, opening up new research directions for future work to build on.
>
> >W1: The proposed safety data selection rule can be viewed as a relatively natural heuristic that ranks candidate examples based on the loss gap between an aligned model and a set of misaligned models.
>
> We agree that our data selection rule is a natural heuristic, and consider this a strength rather than a weakness.
>
> >W2: Computational overhead
>
> We assume the scenario of an API model provider who offers fine-tuning their model $M$ on fine-tuning data $\mathcal{D}_f$ into a fine-tuned model $M_f$, i.e. we expect the same model $M$ to be fine-tuned many times. However, the cost of data selection in interleaving++ is a one-time cost, namely, (1 + n_misaligned_models) forward passes over the candidate interleaving dataset. This is a relatively low one-time cost that will furthermore quickly amortize over many fine-tuning runs in a production setting. The cost of filtering refusals is negligible. We will clarify this in the camera-ready version.
>
> For example, consider a 7B model quantized to 8 bits, n_misaligned_models = 3, 100 finetuning runs with 6000 finetuning data points each and 10 000 candidate interleaving data points to be scored on a single A100 GPU. We estimate that loss data selection takes ~0.7 GPU hours, while the fine-tuning runs take a total of ~80 GPU hours. We assume that fine-tuning takes 9× as much compute as a forward pass due to the extra cost of a backward pass and lower MFU due to gradient and optimizer overhead. The overhead can in general be computed as
>
> overhead = (1 + n_misaligned_models) * data_ratio / (train_to_inference_cost_ratio * n_finetuning_runs)
>
> The extra overhead from interleaving++ versus interleaving is thus on the order of 1% or less in a realistic production setting. We will include this cost analysis in the appendix.
>
> >W3: Comparison with coresets
>
> The coreset selection method is an interesting approach that could provide some benefit in our usecase, by allowing us to replace the interleaving set with a coreset while potentially retaining performance.
>
> However, the core issue is that coreset selection and Interleaving++ solve fundamentally different problems. Coreset methods aim to find a small subset $\mathcal{S}$ of a dataset $\mathcal{D}$ such that training on $\mathcal{S}$ approximates training on all of $\mathcal{D}$—the objective is data efficiency for the same task. In our setting, we are not trying to approximate training on the full WildGuardMix dataset. Rather, we try to find a small subset of examples that maximally counteract the alignment-eroding effect of a different dataset (the fine-tuning data). The optimization target is therefore not "approximate the full safety dataset" but "prevent persona shift induced by $\mathcal{D}_f$."
>
> This makes the methodological lineage much closer to Moore & Lewis (2010) style cross-entropy difference data selection from the domain adaptation literature, which we already cite.
>
> The key distinction there is that [2] selects from the training set itself based on per-example training dynamics, whereas we score examples from an external pool using a set of frozen reference models. There's no trajectory to compute over because the safety data isn't part of the original fine-tuning run.
>
> There's also a practical constraint: in the API provider setting, the safety pool and its scoring should ideally be precomputed before any customer submits fine-tuning data. Coreset methods that require access to the training dynamics or gradients of the actual fine-tuning run would add substantial latency and computational cost to every API fine-tuning job. Our method scores the candidate pool once using a set of reference misaligned models, and the ranking can be reused across customers, amortizing quickly.
> We will mention coresets in the related work section and clarify how our approach differs in terms of objectives and intended application setting.

---

> > ### Author Rebuttal · Reviewer_C18y · 2026-04-01
> >
> > Thank you for the detailed rebuttal. Most of my concerns have been addressed, and I will raise my score.

---

> > > ### Author Response · Authors · 2026-04-02
> > >
> > > We're glad that our rebuttal has addressed your concerns. Your review has made our paper substantially better. Thanks for raising the score.

---

### Official Review · Reviewer_Rcpt · 2026-03-12

**Soundness:** 3
**Presentation:** 4
**Significance:** 3
**Originality:** 3
**Overall Recommendation:** 5
**Confidence:** 3

**Summary:**

The paper presents an empirical study of methods that can defend against emergent misalignment (EMA), the phenomenon where models become egregiously misaligned after being fine-tuned on benign but out-of-distribution examples, such as incorrect code. The paper studies various measures that a fine-tuning provider could impose to avoid EMA, such as KL regularization, periodic re-training on safe data, and persona steering vectors. The authors show that their Interleaving++ method performs the best at reducing EMA.

**Compliance With Llm Reviewing Policy:**

Affirmed.

**Final Justification:**

As described in my reply to the authors, this is a good paper, the rebuttal addressed my questions and I recommend acceptance

**Key Questions For Authors:**

+ Could you provide error bars for your results in table 2?
+ How does the choice of misaligned model vs the model under training interact with the effectiveness of Interleaving++? E.g. if I try to train a model to be emergently misaligned in a particular narrow domain (perhaps by preemptively mixing in benign examples in e.g. code), will a generically EMA model still work well for interleaving++?

**Limitations:**

yes

**Strengths And Weaknesses:**

# Strengths
+ The problem is clearly an important and timely one--EMA is a problem for finetuning providers, since adversarial users can produce jailbroken models without necessarily inputting any explicilty malicious finetuning data.
+ The paper has a fairly comprehensive empirical evaluation, with four distinct classes of regularization methods across four EMA domains and two benign benchmarks, and extensive hyperparameter ablations. The results are correspondingly broad: no single method dominates across all criteria, and the paper is clear about the resulting trade-offs.
+ The Interleaving++ data selection method is a nice contribution. The idea of selecting safety data by maximizing the perplexity gap between aligned and misaligned models is straightforward but effective. The implementation details such as filtering refusal-heavy examples show that the implementation is 'deployment ready'.

# Weaknesses
+ There are no error bars for any of the results in table 2. For many of the results this is not a huge issue, since the effect size is quite large, but for some of the comparisons this makes it hard to tell what is a real effect and what is just variance with the small evaluation set (24 questions and LLM-as-a-judge).
+ It's not clear what models are used for the misaligned model in the interleaving++ perplexity estimation. As far as I can tell, there isn't a discussion of how close the deliberately misaligned models have to be to the model being trained, or the extent to which providers have to guess what sort of models the attackers might try to generate--for instance, if the targeted EMA domain is very narrow, will this procedure still work?

---

> ### Author Rebuttal · Authors · 2026-03-30
>
> We thank the reviewer for the constructive feedback. Below we provide responses to each of the raised concerns.
>
> >W1, Q1: There are no error bars for any of the results in table 2.
>
> Indeed, due to computational cost and time constraints we only ran a single run with each combination of hyperparameters. While repeating the fine-tuning process is relatively cheap, the evaluation process via LLM-as-a-judge with powerful closed models incurs significant monetary cost: Each question in the evaluation set is evaluated 100 times, so the actual size of the evaluation for each data point is 2400 samples. As the reviewer acknowledges, the mitigation effect sizes and disparities between mitigation methods are large and consistent across datasets.
>
> >W2: It's not clear what models are used for the misaligned model in the interleaving++ perplexity estimation.
>
> We assume the scenario of an API model provider who offers fine-tuning their model $M$ on fine-tuning data $\mathcal{D}_f$ into a fine-tuned model $M_f$. As a defense, for each of the 4 EM-inducing datasets, we fine-tune $M$ into misaligned models $M_1, M_2, M_3$ trained on the 3 other datasets (i.e. NOT including the evaluation dataset) and average their scores.
>
> >W2: there isn't a discussion of how close the deliberately misaligned models have to be to the model being trained [...] for instance, if the targeted EMA domain is very narrow, will this procedure still work?
>
> >Q2: How does the choice of misaligned model vs the model under training interact with the effectiveness of Interleaving++?
>
> Note that $M_f$ and $M_1, M_2, M_3$ all result from the same base model $M$, so they are necessarily similar. In particular, to answer the concern about narrow domain training data, we note that interleaving++ is effective even when applied to the code domain, while the scoring models for this domain are trained on medical/legal/security advice. The data that is mixed in is from a general domain. Therefore, our experiments suggest that interleaving++ works well even when fine-tuning on a narrow domain divergent from the scorer models. We will emphasize this point more clearly in the camera-ready version.

---

> > ### Author Rebuttal · Reviewer_Rcpt · 2026-04-02
> >
> > Thanks for the rebuttal, I appreciate how resource limitations can make exhaustive experimentation difficult--you have answered my other questions.
> >
> > I don't think this paper is quite at a 'best paper' level so won't raise my score to 6, but it is certainly deserving of being accepted and I recommend as such to the AC.

---

> > > ### Author Response · Authors · 2026-04-04
> > >
> > > We're glad that our rebuttal has addressed your concerns. Your review has helped us make our paper substantially better. Thanks for the positive review and for recommending acceptance.

---

### Decision · Program_Chairs · 2026-04-30

**Decision:**

Accept (regular)

**Comment:**

I thank the authors for their interesting work!  This paper studies in-training defenses against EMA that could be practical for API finetuning-as-a-service providers.  The paper ultimately suggests “selecting interleaving data by the perplexity gap between aligned and misaligned models”.  Reviewers agreed that the paper conducts a thorough investigation of an important problem and that interleaving++ is a valuable contribution.  Reviewers did point out several potential weaknesses including missing comparisons to other selection baselines, limited conceptual insights beyond empirical trends, missing statistical analysis which could be important for a largely empirical paper, and computational costs.  The authors largely addressed this feedback in the rebuttals, and the reviewers were generally satisfied by the points made by the authors.  All in all, I tend towards acceptance here.